# Evidence for increased parallel information transmission in human brain networks compared to macaques and male mice

Alessandra Griffa [1,2,3] ✉, Mathieu Mach[2], Julien Dedelley[2], Daniel Gutierrez-Barragan[4], Alessandro Gozzi [4], Gilles Allali[1], Joanes Grandjean[5,6], Dimitri Van De Ville [2,3] & Enrico Amico [2,3] ✉

Brain communication, defined as information transmission through white-matter connections, is at the foundation of the brain's computational capacities that subtend almost all aspects of behavior: from sensory perception shared across mammalian species, to complex cognitive functions in humans. How did communication strategies in macroscale brain networks adapt across evolution to accomplish increasingly complex functions? By applying a graph- and information-theory approach to assess information-related pathways in male mouse, macaque and human brains, we show a brain communication gap between selective information transmission in non-human mammals, where brain regions share information through single polysynaptic pathways, and parallel information transmission in humans, where regions share information through multiple parallel pathways. In humans, parallel transmission acts as a major connector between unimodal and transmodal systems. The layout of information-related pathways is unique to individuals across different mammalian species, pointing at the individual-level specificity of information routing architecture. Our work provides evidence that different communication patterns are tied to the evolution of mammalian brain networks.

Understanding how brain function can be supported by patterns of neural signaling through its structural backbone is one of the enduring challenges of network and cognitive neuroscience[1]. The brain is effectively a complex system, a network of neural units interacting at multiple spatial and temporal scales through the white-matter wiring[2,3]. Information transmission through structural connections, which can be defined as brain communication[1], gives rise to macroscale patterns of synchronous activity –or functional connectivity– between remote areas of the brain. Communication processes are at the foundation of the brain's computational capacities that subtend almost all aspects of behavior, from sensory perception and motor

functions shared across mammalian species, to complex human functions including higher-level cognition[4]. From an evolutionary perspective, high communication efficiency at minimal structural wiring cost has long been recognized as a fundamental attribute constraining the evolution of neural systems[5–7]. Yet, quantitative and comparative assessments of macroscale communication processes in mammalian brain networks at different phylogenetic leaves are lacking[1,8].

Systems-level neuroscience has made different attempts to map brain communication as interrelated patterns of macroscale structural and functional brain connectivity, highlighting strikingly complex

[1]Leenaards Memory Center, Lausanne University Hospital and University of Lausanne, Lausanne, Switzerland. [2]Medical Image Processing Laboratory, Neuro-X Institute, École Polytechnique Fédérale De Lausanne (EPFL), Geneva, Switzerland. [3]Department of Radiology and Medical Informatics, University of Geneva, Geneva, Switzerland. [4]Functional Neuroimaging Laboratory, Center for Neuroscience and Cognitive systems, Istituto Italiano di Tecnologia, Rovereto, Italy. [5]Department of Medical Imaging, Radboud University Medical Center, 6525 GA Nijmegen, The Netherlands. [6]Donders Institute for Brain, Cognition and Behaviour, Radboud University Medical Center, 6525 EN Nijmegen, The Netherlands. ✉e-mail: alessandra.griffa@chuv.ch; enrico.amico@epfl.ch

structure-function interdependencies[9]. Structurally connected region pairs tend to have stronger functional connectivity than disconnected pairs[10,11], indicating the presence of monosynaptic interactions[12]. Nonetheless, direct structural connections alone are not able to explain most of the dynamic functional repertoire observed in a functioning brain[13]. Beyond monosynaptic interactions, functional connectivity between remote brain areas is likely to emerge from more complex, higher-order communication mechanisms that involve larger groups of neural elements and their structural interconnections, possibly through polysynaptic (multi-step) routing of neural information[1,14,15]. Brain networks of several mammalian and simpler species have short structural path length[16,17] at the price of a relatively high wiring cost[5], suggesting that polysynaptic shortest paths contribute to efficient communication in brain networks and have been selected throughout evolution despite their high wiring cost. Yet, models of selective communication through shortest paths only explain a limited portion of functional connectivity[10] and exclude a large fraction of brain network connections and near-optimal alternative paths from the communication process[18], pointing out the relevance of multiple paths to brain communication models[1,19,20]. Indeed, in many real-world systems, information transmission unfolds through numerous alternative pathways according to parallel communication schemes[21]. In the brain, parallel communication may increase transmission fidelity, robustness and resilience to brain damage[1] while achieving a reasonable trade-off between communication efficiency and metabolic expenditure[22]. Moreover, multiple communication channels may be used together or separately at different moments in time to support changing internal and external representations and complex functions including higher-order cognition[23]. In this sense, a compelling hypothesis is that parallel communication may have evolved across mammalian evolution to support cognitive tasks of increasing complexity. Nonetheless, the information transmission mechanisms implemented in mammalian brain networks and, particularly, the relative contribution of single-pathway ('selective') versus multiple-pathway ('parallel') information transmission are, to date, largely unknown.

Comparative neuroimaging provides instruments to understand the emergence of function across evolution[17,24]. Evidence of similarities between neural systems in different species are assumed to reflect common organizational principles and functions that may be evolutionarily preserved. In contrast, regions showing the greatest changes between humans and other species may account for features of cognition unique to humans. It has been shown that the overall topology of the structural and functional brain networks is preserved across evolution[16,17] despite large variations in brain size and cortical expansion[25]. However, differences in local connectivity patterns and functional dynamics exist[26–28]. Functional patterns extending beyond pairwise-connected regions, possibly supported by polysynaptic paths, have been identified across different mammals[29]. Moreover, functional patterns untethered from structure are dominant in cortical areas that underwent larger evolutionary expansion across primates, suggesting a relation between local information transmission mechanisms and evolution[25,30,31]. These considerations question whether macroscale polysynaptic communication mechanisms are preserved across species or, contrarily, differ between phylogenetic leaves. In particular, was there a shift from single-pathway (selective) to multiple-pathway (parallel) communication across evolution to support increasingly complex brain functions? This question does not have trivial answers and demands for new ways of assessing brain communication across different species.

Here we propose an approach, rooted in graph and information theory, to model polysynaptic information transmission in macroscale brain networks. Taking advantage of structural and functional connectivity information extracted from multimodal brain data (i.e., functional MRI, diffusion MRI, tract tracing), we explore the intricate architecture of information-related pathways in the mouse, monkey and human connectomes. We employ information-theoretical principles[22,32,33] to identify the structural paths supporting information transmission in different neural systems, and measure the level of selective and parallel communication across the different species. We report a brain communication gap between humans and non-human mammals, with predominant selective information routing in mice and macaques, morphing into more complex communication patterns in human brains. Parallel communication strategies appear to have acted as a major connector of unimodal (sensory, attentional) and transmodal (fronto-parietal, default mode) areas in the human brain, possibly contributing to the evolution of more complex cognitive functions in humans. Notably, we also found that information-related pathways are highly specific to individuals across the different mammalian species. Our results link the complexity of macroscale brain communication dynamics inferred from in vivo data to the gap between human and non-human mammalian lineages. These findings pave the way to a deeper understanding of how brain communication and its relationship to function have evolved across species.

## Results

We introduce a graph- and information-theory framework to model information-related pathways in brain networks and investigate their evolution in three mammalian species: mice, monkeys, and humans (Fig. 1). These species represent distinct mammalian lineages and include animal models (mice, monkeys) often used in translational research. We aimed to investigate two general aspects of brain communication dynamics. First, due to the noisy nature of neural signaling and under a relay information transmission model, neural messages transmitted through the structural brain network can keep at most the same amount of information present at the source region[22,32]. This holds true for many communication systems where the information content tends to decay as one moves away from the information source[34]. Second, in an information transmission process, messages are typically relayed through a set of statistically independent steps[35]; i.e., neural messages do not contain memory of the transmission process itself and communication happens in a Markovian fashion[36]. These two dimensions –information decay and memoryless transmission– are formally summarized by a fundamental principle of information theory, the data processing inequality (DPI)[33], which we here apply to cross-species structural data and fMRI recordings (Methods). Our model answers the following research question: assuming that a source brain region is transmitting information to a destination region, is this exchange of information compatible with a relay communication channel (in the DPI-sense)? To answer this question, we first identify sets of short polysynaptic paths connecting regions pairs in the structural network representing the white matter wiring of the brain (Fig. 1a). These structural paths are made of several hops from cortical region to neighborhood cortical region in the structural connectome. We note that the structural connectomes were estimated with different modalities in the three species (diffusion tractography and/or tract tracing) to cope with the difficulties in estimating diffusion-based connectomes in smaller brains (Methods). Next, we estimate which and how many of those structural paths are truly compatible with Markovian neural information transmission. To this aim, we quantify fMRI-derived mutual information measures along the structural paths (Fig. 1b) to assess the DPI on those paths (Fig. 1c). From here, a parallel communication score can be computed for every pair of brain regions, by counting the number of paths that respect the DPI (Fig. 1d). Parallel communication scores portray a spectrum of communication strategies from selective information transmission, where brain regions selectively exchange information through a single pathway, to parallel information transmission, where regions communicate through multiple, parallel pathways (Fig. 1e). We assessed

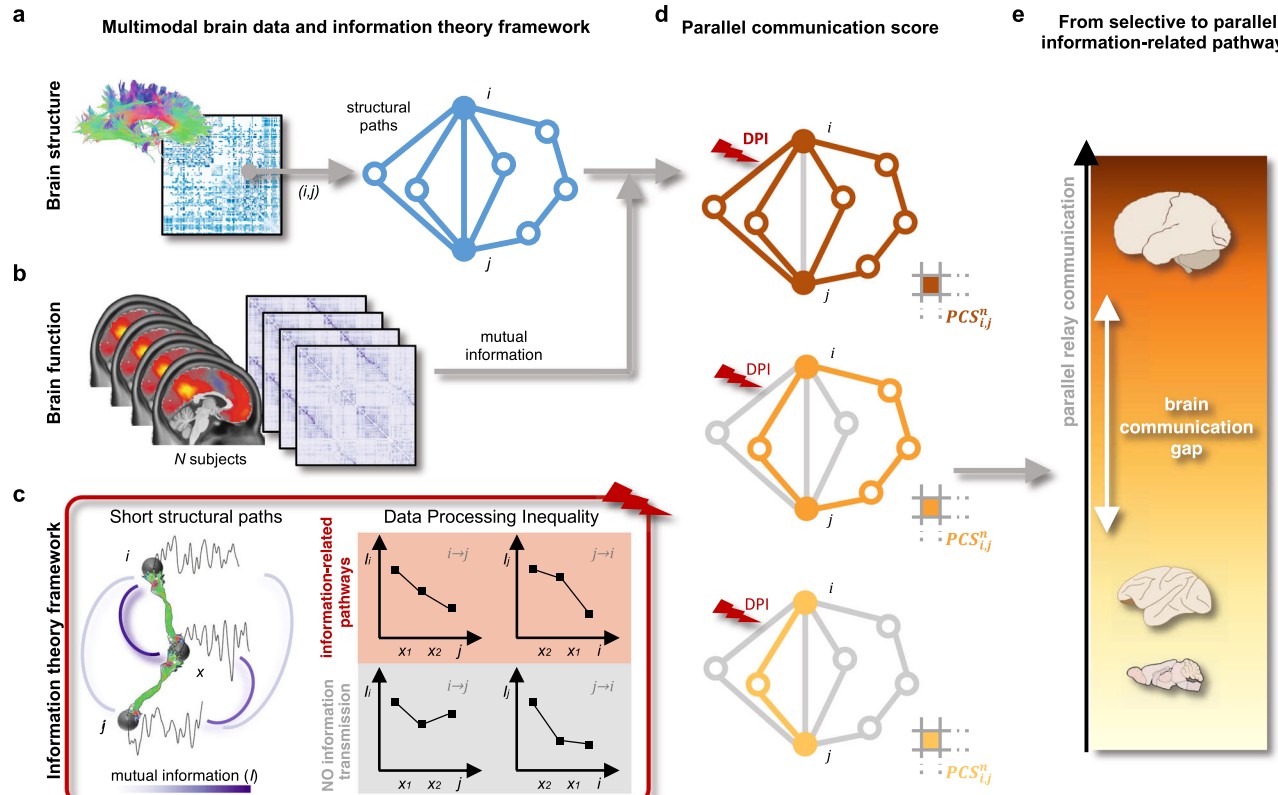

**Fig. 1 | Identifying information-related pathways in macroscale brain networks.**
**a** A weighted and symmetric structural connectivity matrix summarizes the white matter wiring of the brain for each species. For every pair of brain regions *(i, j)*, the 5 shortest structural paths (light blue) connecting the two regions are identified using the *k*-shortest path algorithm[18]. **b** For every (human or non-human) subject, the mutual information between region pairs is computed from *z*-scored regional time series obtained from fMRI recordings. **c** By analyzing the mutual information values along each structural path, the data processing inequality (DPI) is used to assess whether the specific structural path represents an information-related pathway between regions *i* and *j*. Left panel: two brain regions *i, j* are connected by a structural path crossing regions $x_1, x_2$; green lines represent direct structural connections (white matter fibers). Each region is associated with a neural activity-related time series; the amount of information shared by two regions is quantified by their mutual information *I* (darker and thicker arcs indicate stronger *I*). Right panel: a structural path $(i, x_1, x_2, j)$ is labeled as relay information-related pathways if the pairwise mutual information values do not increase along the (undirected) path (first row, red shading); it is not an information-related pathway otherwise (second row, gray shading: $I_{j,i} > I_{x2,i}$). **d** A parallel communication score (PCS) is computed at the individual level (i.e., for every subject *n*) and for every pair of brain regions *i, j* by counting the number of structural paths that serve as information-related pathways between the two regions. **e** Parallel communication scores are investigated across mammalian species, highlighting a spectrum of communication strategies from selective information transmission (light yellow; low PCS), to parallel information transmission (dark brown; high PCS). Particularly, our work highlights a parallel relay communication gap between humans and non-human mammals (macaques, male mice), with humans' brain network communication tailored towards parallel transmission, and macaques and mice towards selective transmission. Brain schematic from scidraw.io[97–99].

parallel communication scores at the individual and group levels, and summarized them for distinct brain systems and single brain regions in comparison to appropriate null models. Finally, we compared the distribution of selective and parallel information transmission across the three different mammalian species.

Data for this study consisted of open-source whole-brain structural connectivity matrices and individual resting-state functional MRI recordings of 100 healthy human subjects, 9 macaque monkeys, and 10 male wild-type mice, all in their young adulthood (Methods, Supplementary Table 1; see below for replication datasets). All main functional data were recorded from non-anesthetized experimental subjects. Group-representative structural connectivity matrices with comparable number of brain regions were derived from diffusion MRI and/or tract tracing data, and weighted by the Euclidean distance between connected regions since, from a neurobiological perspective, neural signal transmission through physically shorter axons is less costly in terms of metabolic resources and may be preferred to longer connections[6] (Supplementary Fig. 1). Individual-level mutual information matrices were computed from fMRI recordings of comparable duration and temporal resolution across species.

## Parallel information transmission in brain networks shows a gap between humans and non-human mammals

Using our approach we found that, in mammalian brains, polysynaptic structural paths are used to relay information in a "Markovian"-specific, sequential processing fashion. For all the three considered species, the whole-brain density of relay information-related pathways (i.e., the percentage of structural paths respecting the DPI) was higher than in a strict null model preserving the structural connectivity architecture and the multivariate statistics of fMRI time series (Methods; two-sided Mann-Whitney *U* test, $p < 10^{-5}$ for all species). This held true when considering either the first shortest path connecting region pairs (mean ± standard deviation across individuals: humans = 39.5 ± 3.8%; macaques = 37.9 ± 5.8%; mice = 40.1 ± 2.4%), or longer paths (humans = 33.0 ± 5.6%; macaques = 29.8 ± 6.3%; mice = 30.4 ± 6.2%), showing that relay information transmission is not limited to the shortest path only (Fig. 2a). Specifically, the communication density levels were comparable for the first and second shortest paths and decreased for longer paths in all the three species. Next, we assessed the amount of parallel information transmission between all brain region pairs. We found that, on average, the parallel communication score (PCS) was larger in humans compared to animals (median [5-,

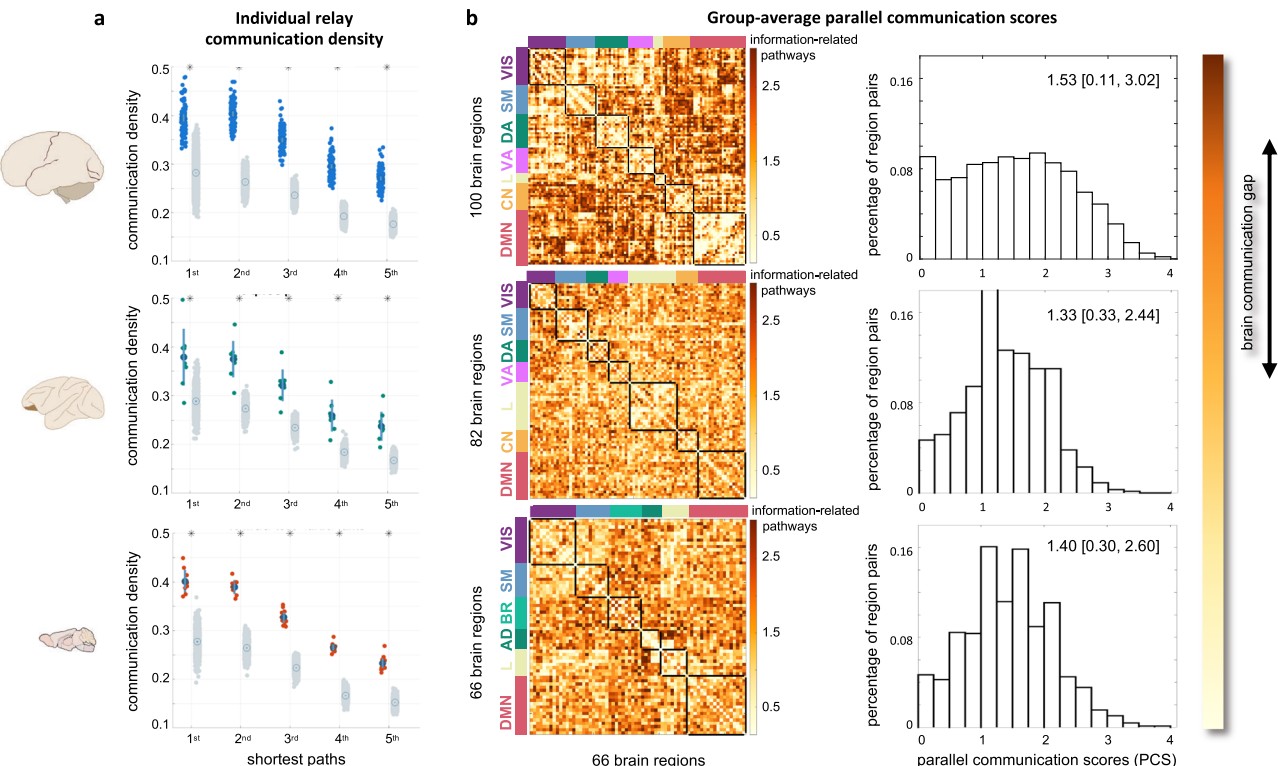

**Fig. 2 | Parallel information transmission gap across mammalian species.** Left: schematic of human, macaque, and mouse brains from scidraw.io[97–99]. Each row in the figure corresponds to one species. **a** Box plots representing the percentage of short structural paths in individual brain networks respecting the data processing inequality (DPI) (communication density). Each colored dot represents an individual (humans: $n = 100$ biologically independent subjects; macaques: $n = 9$ biologically independent subjects; mice: $n = 10$ biologically independent subjects); gray dots represent species-specific null distributions obtained from random shuffling of fMRI time series (Methods); circles and vertical bars indicate mean ± one standard deviation across individuals or randomizations. Paths are grouped according to the 1st up to the 5th shortest path between region pairs, showing that relay communication is not limited to the 1st shortest path only. * indicate $p < 10^{-5}$ for two-sided Mann-Whitney $U$ test between subject and null distributions. **b** Group-

average parallel communication score (PCS) matrices representing PCSs between every pair of brain regions, averaged across individuals. The color scale unit is the number of information-related pathways. For each species, brain regions are organized according to meaningful functional circuits which are highlighted by black squares along the matrices' diagonals and by color-coded bars (Methods). On the right, the histograms of the average PCS scores across region pairs highlight a brain information transmission gap between humans, with higher PCSs and presence of parallel information transmission, and macaques and mice, with lower PCSs and mainly selective information transmission. Median [5-, 95-percentile] PCS values for each species are reported atop each histogram. VIS visual, SM somatomotor, DA dorsal attention, VA ventral attention, L limbic, CN control, DMN default mode, BR barrel, AD auditory networks. Source data are provided as a Source Data file.

95-percentile] across region pairs: humans = 1.52 [0.11, 3.02]; macaques = 1.33 [0.33, 2.44]; mice = 1.40 [0.30, 2.60]). This is particularly evident when considering the long-tailed distribution of human PCSs as compared to the non-human mammals (Fig. 2b). The three species' PCS distributions were pairwise statistically different (two-sample Kolmogorov-Smirnov tests human-macaque: $D_{4950,3321} = 0.180$, $p < 10^{-55}$; human-mouse: $D_{4950,2145} = 0.144$, $p < 10^{-27}$; macaque-mouse: $D_{3321,2145} = 0.076$, $p < 10^{-6}$) and the brain information transmission gap between humans and non-human mammals was consistent when considering replication datasets (two-sided Mann-Whitney $U$ tests, Supplementary Fig. 2). These results indicate that, as the brain evolved from mice and macaques to humans, interareal communication is subserved by parallel transmission (Fig. 2).

**Information transmission architecture is species-dependent and relates to the functional organization of mammalian brains**

When evaluating the spatial localization of the relay information-related pathways, we found that it followed the characteristics of each species' functional cortical architecture (Fig. 3). In lower species the relay (mostly sequential) pathways mainly encompassed unimodal/multimodal regions spanning the barrel and auditory cortices in mice, and the visual cortices in macaques (Fig. 3a). These patterns were preserved in anesthetized animals (replication datasets, Supplementary Figs. 16, 17). In humans we found similar evidence of relay

sequential transmission in unimodal and multimodal areas, but also a high concentration of parallel information-related pathways in transmodal regions including association cortices of the executive-control network and the precuneus of the default mode network (Fig. 3a; note the different color scales across species).

Next, we investigated the communication patterns at the level of region pairs within and between different brain functional systems (Methods; Supplementary Figs. 3, 4). In mice and macaques, the relay pathways mainly connected brain nodes belonging to unimodal systems, particularly barrel and auditory cortices in mice, and visual with somatomotor and attention regions in macaques (Fig. 3b). In humans, stronger (parallel) relay information transmission mainly connected somatosensory and attention regions with executive-control and default mode systems, forming cross-modal parallel streams between unimodal and transmodal regions (outside-diagonal entries in Fig. 3b). Notably, these patterns were stable at the individual subject level. We report in Fig. 3c the amount of relay information transmission (average PCSs) within unimodal systems, within transmodal systems, and between unimodal and transmodal ("cross-modal") systems for each experimental subject and each species. The amount of relay information transmission strongly varied between systems in humans, with cross-modal region pairs presenting the highest amount of relay transmission and transmodal regions the lowest amount (Kruskal-Wallis test, $p < 10^{-44}$), and moderately varied in macaques and mice,

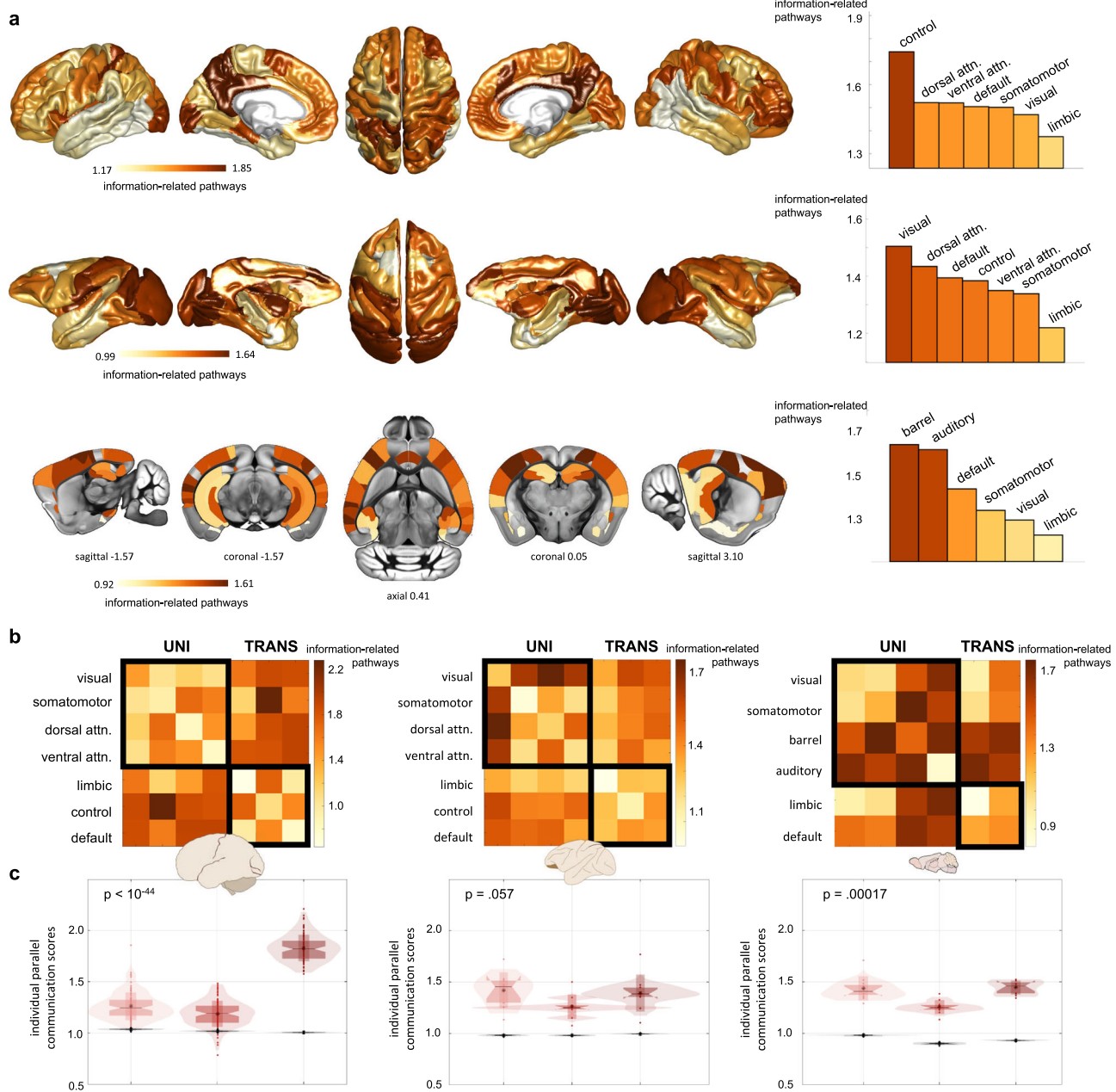

**Fig. 3 | Relay information transmission strategies reflect the functional organization of mammalian brains. a** Cortical distributions of relay information-related pathways, quantified as the average PCS of each brain region with the rest of the brain network (first row: human, fsaverage6 cortical surface; second row: macaque, F99 template; third row: mouse, ABI template). For each species, the light yellow-to-brown colormap is scaled between the 5th and 95th percentiles of the cortical values, with the color scale unit indicating the average number of information-related pathways. On the right, the average nodal PCS scores per brain system are represented in the bar plots. **b** Average PCSs within and between brain systems, for humans, macaques and mice. Brain systems have been organized into unimodal/multimodal regions (upper-left black square) and transmodal regions (lower-right black square). **c** Average PCSs between unimodal systems, between transmodal systems, and between unimodal and transmodal systems (cross-modal communication) for individual experimental subjects. In the box plots, each dot represents a subject (humans: $n = 100$ biologically independent subjects; macaques: $n = 9$ biologically independent subjects; mice: $n = 10$ biologically independent subjects); vertical bars indicate mean ± standard deviation; notch bars indicate median and 1st–3rd quartiles; shaded areas indicate 1st-99th percentiles. Kruskal Wallis p-values for within-species comparisons are reported, testing the null hypothesis that unimodal, transmodal and cross-modal PCS scores originate from the same distribution (humans: $p < 10^{-44}$; macaques: $p = 0.057$; mice: $p = 0.00017$). Values from the null model are reported in the same panels as gray box plots. Brain schematic from scidraw.io[97–99]. Control=executive-control, dorsal attn.=dorsal attention, ventral attn.=ventral attention, uni=unimodal/multimodal, trans=transmodal systems. Source data are provided as a Source Data file.

with transmodal networks consistently presenting the lowest amount of relay transmission (Kruskal-Wallis test, $p = 0.00017$) (Fig. 3c). Across species, relay information transmission within unimodal systems and between unimodal and transmodal systems differed across species (Kruskal-Wallis tests, $H(2) = 20.17$, $p = 0.000042$ and $H(2) = 44.36$, $p < 10^{-9}$, respectively), with humans presenting 30% larger PCSs in cross-modal regions than macaques and mice, independently from the presence of anesthesia in animals (Supplementary Fig. 5). Relay transmission within transmodal systems was relatively stable across species ($H(2) = 6.47$, $p = 0.039$). Taken together, these findings show

that communication strategies are highly heterogeneous across the brain network and are partially preserved across evolution. However, non-human mammals demonstrate more developed selective communication for lower-order processing between unimodal and multimodal regions. Conversely, the human brain is characterized by stronger parallel communication that serves as the main neural communication stream between unimodal and transmodal areas[37,38].

### The layout of information-related pathways is unique to individuals

Our results revealed a link between relay information transmission and the phylogenetic level of mammalian brains. Are the observed communication patterns specific to individual subjects within single species? We addressed this question by exploring the identifiability properties[39,40] of the parallel communication matrices reported in Fig. 2b, across the three different species. To this aim, the fMRI recording of each subject was split into two sections of equal duration. From these, test and retest parallel communication matrices were computed. Note that, at the individual level, the matrices' entries (PCSs) can take integer values between 0 and 5, with 0 indicating no relay communication, 1 indicating perfectly selective information transmission, and 5 strongly parallel information transmission. We quantified the similarity between test and retest data as the percentage of brain regions' pairs with exactly the same PCS (Jaccard similarity index). The individual identifiability through relay communication patterns was then quantified as the success rate (SR), i.e., the percentage of subjects

whose identity was correctly predicted out of the total number of subjects for each species[40]. We found that parallel communication scores allow to identify individual mammals in all the three species, at a level that exceeds chance-level (humans: SR = 87.0%, null = 0.9 ± 1.0%; macaques: SR = 66.7%, null = 10.2 ± 10.0%; mice: SR = 40.0%, null = 10.7 ± 9.8%) (Fig. 4a, b). However, individual identifiability decreased from humans, to macaques, to mice. Intriguingly, the major contribution to individual identifiability was given by brain regions pairs that, on average, tends to communicate through multiple parallel rather than selective pathways. When splitting region pairs into two groups ("low-PCS", "high-PCS") according to group-average parallel communication scores, the success rate obtained from high-PCS values was higher than the one obtained from low-PCS values for humans; no differences were found in macaques and mice (PCS threshold = 1.3; humans: SR_{low-PCS} = 73.0%, SR_{high-PCS} = 85.0%; macaques: SR_{low-PCS} = 66.7%, SR_{high-PCS} = 66.7%; mice: SR_{low-PCS} = 40.0%, SR_{high-PCS} = 40.0%; see Supplementary Table 3 for alternative PCS thresholds) (Fig. 4c, d). Taken together, these data suggest that, within the inherent constraints of each species, individual subjects may be characterized by distinct communication patterns to relay neural information through the brain network, particularly when considering higher-order communication mechanisms such as parallel communication in humans.

### Robustness, sensitivity, and replication analyses

To ensure the validity of our results, we asked whether parallel communication scores and their cross-species gap could be explained by

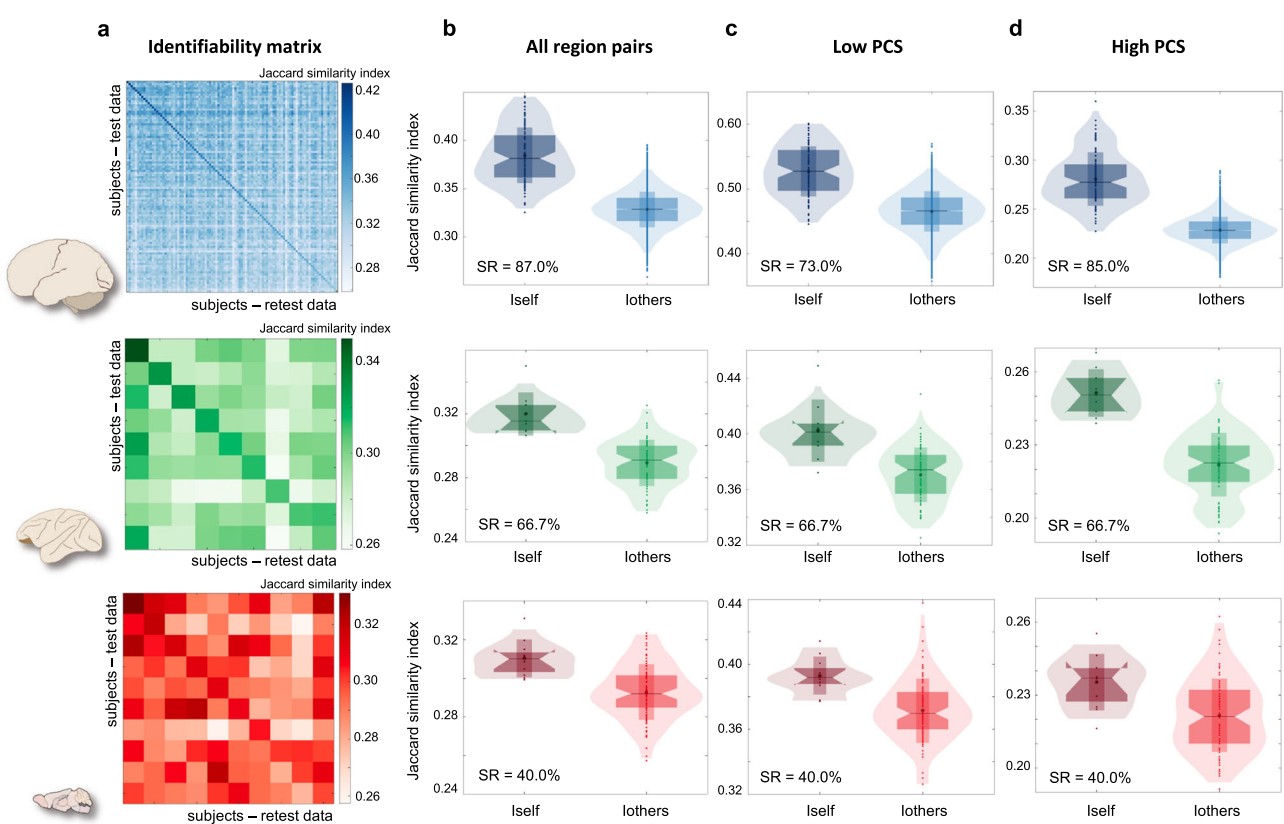

**Fig. 4 | Information-related pathways are unique to individuals.** Left: schematic of human, macaque, and mouse brains from scidraw.io[97–99]. Each row in the figure corresponds to one species. **a** Identifiability matrices for the three species, reporting subjects' similarities between test (rows) and retest (columns) parallel communication score (PCS) data. Humans: *n* = 100 biologically independent subjects; macaques: *n* = 9 biologically independent animals; mice: *n* = 10 biologically independent animals. Test-retest similarity was quantified with the Jaccard similarity index. **b** Box plots representing self-similarity (Iself, diagonal entries of the identifiability matrix) and others-similarity (Iothers, out-diagonal entries of the identifiability matrix) values. **c** Self- and others-similarity values when considering only region pairs with low parallel communication scores (PCS ≤ 1.3 on average). **d** Self- and others-similarity values when considering only region pairs with high parallel communication scores (PCS > 1.3 on average). Test–retest datasets were obtained by splitting into two sections the fMRI recording of each subject. The success rate (SR) for subjects' identification is reported for each pair of box plots. In the box plots, vertical bars indicate mean ± standard deviation; notch bars indicate median and 1st–3rd quartiles; shaded areas indicate 1st–99th percentiles. Source data are provided as a Source Data file.

the signal-to-noise ratio (SNR) and multivariate statistical properties of fMRI recordings alone. To this aim, we constructed null distributions of parallel communication scores for each species, by randomly shuffling the fMRI time series across brain regions and computing surrogate PCSs on the original structural connectome architecture ($n$ = 3000) (Methods, Supplementary Fig. 6). By z-scoring the real PCS scores with respect to the null distributions and applying appropriate statistical thresholds, we show that our findings do not trivially derive from structural connectivity and fMRI time series' SNR or statistical properties alone. In particular, the gap of increasing parallel communication from mice and macaques to humans (Fig. 2, Supplementary Figs. 7, 8, 9) and the cortical topographies of parallel communication density across species (Fig. 3, Supplementary Fig. 10) remained significantly different from the null ones. In addition, we did not find correlations between individual nodal SNR of fMRI time series and nodal PCS scores (Supplementary Fig. 11), suggesting that differences in scanning protocols do not drive cross-species parallel communication gap and individual identifiability results.

Parallel communication scores and unimodal-transmodal variations in parallel communication were not explained by the different structural architectures and brain spatial embedding of the investigated species, nor by the spatial autocorrelation of the mutual information values with respect to the underlying distance-weighted structural connectivity matrix. To test these aspects, we constructed two additional null models. In the first one we populated the network nodes with iid Gaussian noise with mean 0, variance 1, and the same number of time points as in experimental data. This scenario represents the case of absent communication between brain regions interconnected through the real structural connectome. In the second model we randomly permuted the mutual information values of individual subjects while preserving their spatial autocorrelation with respect to the underlying distance-weighted structural connectivity matrix, similarly to the method proposed in ref. 41 (see Methods). Parallel communication densities and PCS scores (group-average and individual-level PCSs of unimodal, transmodal and cross-modal communication streams) were larger than expected based on the two null models (Supplementary Figs. 19, 20).

Even though white matter volume increases from mice, to macaques, to humans, which may increase structural paths' count and thus opportunities for parallel processing, there was no meaningful cross-species difference in structural connectivity density, i.e., relative number of white matter connections (Supplementary Fig. 1; note that connectivity density and paths' count depend on the methodological choices for white matter connectivity mapping). Moreover, individual PCS scores did not depend on connectivity density (Supplementary Fig. 12). There was no difference between human and macaques in the k-shortest path length as quantified by the number of hops along the structural paths (mean (standard deviation) over all region pairs, human: 2.43 (0.58); macaque: 2.43 (0.59); two-sided Mann-Whitney $U$ test, $p$ = 0.98). However, the mouse had on average slightly longer k-shortest paths (2.48 (0.60)) than both the human ($p < 10^{-3}$) and the macaque ($p < 10^{-3}$). Considering the gap in parallel communication between humans and non-human mammals (both macaques and mice), we can rule out that cross-species differences in parallel communication are driven by k-shortest path lengths.

Next, we investigated whether results were sensitive to some methodological choices, including number of brain regions (brain parcellation), fMRI time series length, and number of subjects. We found that our results are robust to these factors. In humans, PCS scores, their cortical topographies and unimodal-transmodal differences were comparable when subdividing the cortex into 100 or 200 regions of interest (Supplementary Figs. 13, 14, 18). However, we observed lower PCS scores when using a finer-grain parcellation with 400 regions (Supplementary Fig. 13). This is expected since structural connectivity and structure-function relationship have been shown to

vary with the number of brain regions[42]. PCS scores tended to increase for longer fMRI time series, but this effect did not impact the corss-species parallel communication gap nor the PCS cortical topographies (Supplementary Figs. 13, 14), and it could relate to the improved reliability of functional connectivity estimation for longer scan lengths[43]. Whenever possible, data from the three species were matched both in the number of brain regions and fMRI scan duration. The effect of the number of subjects on PCS scores was minor (Supplementary Figs. 14, 15, 18).

We assessed the replicability of our findings by analyzing a total of six distinct datasets (Methods). We found that the 2, 15 PCS cortical topographies (Fig. 3, Supplementary Figs. 16, 17), the cross-species parallel communication gap (Supplementary Fig. 2), and the parallel communication differences between unimodal and transmodal regions across species (Fig. 3, Supplementary Fig. 5) were consistent when considering alternative datasets and when comparing awake and anesthetized macaques or mice (Supplementary Figs. 5, 16, 17). However, the overall amount of relay communication was slightly larger in non-anesthetized mice compared to one of the two anesthetized datasets (m-AD3) in both transmodal and cross-modal systems (Supplementary Fig. 5, central and bottom rows). These state-dependent differences were smaller than the ones observed between mice or macaques, and humans.

Finally, we investigated the relationship between parallel communication scores and brain structural and functional connectivity, as assessed with tractography and tract tracing data weighted by the Euclidean distance between regions pairs, and with mutual information values between fMRI time series. The relationship between the PCS and (i) the number of structural connections (structural degree), (ii) the average length of structural connections, and (iii) the overall functional connectivity (functional strength) of individual brain regions were assessed with a multiple regression model including the PCS as dependent variable, and the species and the three nodal connectivity measures as independent variables. There was a significant effect of the species, confirming that PCSs in macaques and mice are smaller than in humans. We found that all three connectivity measures explained significant variance of parallel communication scores (Bonferroni-corrected $p$ values $< 10^{-7}$; Supplementary Table 2; proportion of explained PCS variance (model's R-squared) = 0.42). These results indicate that multiple aspects of the connectome architecture shape (but do not completely determine) communication patterns. Mirroring Fig. 3c, we determined unimodal, transmodal and cross-modal connectome features (density of structural connections, average length of structural connections, and individual-level functional connectivity within the different brain systems) for each species. The observed distributions indicate that the cross-species gap, with strong parallel communication streams between unimodal and transmodal ("cross-modal") areas in humans (Fig. 3c), cannot be explained by structural and functional connectivity features alone (Supplementary Fig. 21).

## Discussion

How networked neural elements intercommunicate at the systems level, ultimately giving rise to brain function, stands as one of the most intriguing and unsolved questions of modern neurosciences. In vivo measurements of brain structure and activity are providing us with windows of opportunities for modeling communication in brain networks, across different animal species. We propose here to bring a piece to this puzzle, by investigating the link between a relay communication model in large-scale brain networks, on the one side, and the phylogenetic level of mammals' brain functions, on the other. By introducing a graph- and information-theory approach to approximate relay information-related pathways in brain networks, we provide compelling evidence that this link exists, and that different communication patterns are tied to the phylogenetic level of mammalian brain

networks along two main organizational principles. The first principle is the parallel communication gap between humans and non-human mammals, with predominantly parallel information transmission in humans and selective information transmission in macaques and mice. The second principle involves the development of cross-system communication through parallel pathways that connect together functionally specialized brain regions (i.e., somatomotor, visual) with transmodal ones (i.e., fronto-parietal, default mode).

Specialized, unimodal brain systems are organized as serial, hierarchical streams where raw sensory information is relayed through stepwise progressive circuits to guide attention and direct actions[25,37]. Consistent with this hierarchical polarity, we found that unimodal regions are mainly characterized by selective information transmission through single pathways, as quantified by low parallel communication scores. This held true for all the investigated mammalian species, suggesting that unimodal selective information transmission is phylogenetically preserved. On the other hand, transmodal regions present a more complex and less understood organization. Back in 1998, Mesulam hypothesized that "the flow of information for intermediary [transmodal] processing displays patterns consistent with parallel and re-entrant processing"[37]. Our findings consolidate this view by showing that information transmission between unimodal and transmodal regions evolved from selective to parallel streams from non-human mammals to humans, who display more advanced cognitive abilities. Parallel communication could therefore represent a more complex form of information transmission beyond hierarchical processing, which might support integration of perceptual modalities into more complex textures of cognition. Yet, it is important to remark that macaques and mice do not occupy the same part of the mammalian evolutionary tree and that future work, eventually including advanced models of brain communication beyond relay information transmission, should delve deeper into cross-species variations of communication patterns.

Which evolutionary mechanisms may have promoted a higher involvement of parallel communication in humans? According to the tethering hypothesis proposed by Buckner and Krienen[25], the fast cortical expansion of transmodal regions in humans led to the untethering of these regions from developmental anchor points. In parallel, humans exhibit a protracted development of white matter connections over childhood and a progressive structure-function untethering in transmodal regions compared to other primates[31]. Cortical expansion and developmental trajectories in humans may have therefore allowed transmodal regions to develop unique cytoarchitectonic[44] and connectional[26] fingerprints, unbounded from the more rigid hierarchical architecture of unimodal systems[38]. The same processes may have also favored the development of new information transmission pathways (parallel communication) to bridge hierarchical unimodal and distributed transmodal regions. Indeed, in humans we observed the largest parallel communication scores in regions that underwent the largest cortical expansion across evolution and the largest changes of structure-function coupling across development, including fronto-parietal association cortices and precunei[35]. Consistently, it has been shown that the level of structure-function coupling in humans is highly heterogeneous across the cortical mantle and reaches a minimum in transmodal regions at adulthood, which may be critical for the maturation of complex cognitive functions[31,45]. In addition, parallel information transmission may be functional to specific processing needs of unimodal-transmodal communication. Recent computational studies suggest that brain regions with largest allometric scaling privilege fidelity rather than compression of incoming signals from unimodal areas[22]. High-fidelity information transmission may be achieved through parallel streaming of redundant signals, expression of a more resilient communication process.

Our results show that parallel communication also contributes to the individual specificity of communication strategies in brain networks. Selective and parallel information transmission allowed identifying subjects in a group with significant accuracy, across different mammalian species. This indicates that the individual layout of relay information-related pathways constitutes an important fingerprint of brain organization, and that this fingerprint is present even in non-human mammals. In humans, brain regions that tend to communicate through parallel rather than selective streams, including transmodal regions, provided the largest contribution to subject identifiability. Consistently, fMRI activity of association and default mode cortices displays larger inter-individual variability in human and non-human primates compared to lower-order regions[40,46,47]. The role of transmodal cortices in individual identifiability is consistent with their protracted neurodevelopment and role in higher-order cognition, and it could partially explain the identifiability gradient observed from humans to macaques, to mice, with mice displaying lower identifiability. However, the identifiability gradients may also be explained by a larger homogeneity among laboratory animals compared to human samples in terms of genetic pedigree and environmental conditions. Subject specificity of communication patterns was assessed on relatively short time series (250–300 time points). Recent studies have shown that one does not need long fMRI scans to achieve high test-retest reliability[39,48]. Based on this recent evidence, it appears that ~100 fMRI volumes would be enough to achieve good success rates and identifiability scores in humans. Furthermore, one common problem in cross-species studies is that it is usually very difficult to acquire test/retest sessions in the different cohorts. One potential workaround to this issue might be to cut the resting state time series in half (as originally proposed in[39]). This has the benefit of removing the scanner and acquisition noise, which is usually a major confound in connectome identification[49]. It is quite established that between-scanner or multi-site acquisitions and their subsequent analyses include the scanner-dependent variability that can mask true underlying changes in brain structure and function. In fact, it is known that even when using identical (let alone "comparable") imaging sequences and parameters, potential site-dependent differences might arise due to a range of physical variables, including field inhomogeneities, transmit and receive coil configurations, system stability, system maintenance, scanner drift over time and many other[50–52]. However, it comes at the cost of looking at within-session fingerprinting, hence focusing more on the temporal stability aspect of the communication pattern, rather than on standard between-session identification. Nonetheless, it is noteworthy that humans could be better identified than non-human mammals solely on the basis of their (within-scan) parallel communication profiles. Future studies should explore how the results change when considering multiple and longer sessions, whenever available, to estimate cross-species communication fingerprints. Finally, the analyses reported in Fig. 4 showed that the parallel communication score is specific to individuals: This is particularly interesting considering that the PCS, which can take integer values between 0 and 5, is a highly compressed measure with respect to functional connectomes. Yet, our analyses do not allow us to draw any conclusions on which brain dimension (structural connectivity, functional connectivity, parallel communication among others) is the most subject-specific or most appropriate for a fingerprinting analysis per se.

Importantly, the brain communication gap from selective to parallel information transmission and the cortical topographies of parallel communication patterns were not explained by cross-species differences of structural connectivity architecture, statistical properties of fMRI data, or conscious (i.e., awake vs anesthetized) state. In keeping with previous studies[16], we found that the overall distribution of short structural path lengths was similar between species, with comparable amounts of 2-step, 3-step, and 4-step paths. Relative cross-species differences of parallel communication were unchanged when contrasting data with respect to different species-specific null models which preserve multivariate fMRI statistics, spatial autocorrelation of

mutual information with respect to the underlying distance-weighted structural connectivity matrix, and/or the structural connectome architecture. Moreover, parallel communication scores did not depend on structural connectome density or SNR of fMRI time series. In line with previous work[1,19], both structural and functional connectome features contributed to shape communication patterns. Nonetheless, parallel communication differences between unimodal and transmodal areas were not explained by the structural and functional connectome architecture alone. When considering fMRI data from awake and anesthetized animals, we found similar cortical distributions of parallel communication patterns, with unimodal regions dominated by selective information processing. This finding is in line with the observation that resting state networks are globally preserved in conscious and unconscious states[29,53]. In general, non-anesthetized animals seem to have a larger amount of relay communication than anesthetized ones. A shift of the communication regime toward more abundant and (partially) parallelized polysynaptic information transmission may support functional integration, inter-network cross-talk, and rich functional repertoires departing from the underlying monosynaptic connectivity constraints, which have been repeatedly observed in awake primates and mice compared to the anesthetized ones[29,53,54]. Nevertheless, the minor anesthesia effects found in this work may be affected by other factors including differences in acquisition protocols and type of anesthesia. For example, different anesthetics have been shown to have different modulatory effects on fMRI brain activity[55]. Our results do not allow us to draw major conclusions on the effect of anesthesia on brain communication processes and further investigation is therefore needed. Particularly, it will be interesting to investigate functional recordings from the same experimental subjects, assessed before and under the effect of anesthesia.

Several higher-order communication models have been proposed to explain integration of information between multiple brain network elements[1]. Nonetheless, the exact polysynaptic communication strategies underlying macroscale neural signaling remain unclear. Intriguingly, brain communication models mostly rely on the assumption of memoryless (Markovian) information transmission[56]. This hypothesis is pervasive in network neuroscience[36] but has never been formally probed in the brain. Our work adds to the field by introducing a framework that explicitly models relay information-related pathways from multimodal brain data, in a way that is grounded in fundamental information-theoretic principles. The framework models memoryless information transmission in brain networks by introducing an empirical way to assess deviations from Markovity through the data processing inequality[32]. Our results show that Markovian communication is consistent with brain data of different mammalian species, is not limited to the shortest structural path, but involves multiple and less optimal structural paths in a way that is species-dependent and consistent with the phylogenetic level of the investigated species.

Polysynaptic memoryless information transmission is a simple model of communication. There is no reason to assume that macroscale neural communication is limited to such a particular form. Brain network hierarchies may confer neural signals a memory of the regions previously visited along a path, thus modifying neural communication pathways in a context-dependent manner[56]. This process would result in non-Markovian communication regimes. The brain may also be modeled through complex multi-object interactions not attributable to information transmission alone, such as synergistic or modulatory behaviors between multiple brain regions; feedback loops; local transformation (non-linear processing) of information[23,36,57,58]. Biologically, these more complex communication patterns may shape important features of the mammalian brain, such as cortical temporal hierarchies[59,60] or receptive time windows for attentional processes[61], and are worthy to investigate in future studies. Previous work showing a strong spatial heterogeneity of structure-function coupling across the cortical mantle has suggested that brain communication mechanisms

may be multiplexed, with multiple protocols operating in parallel[45]. The framework proposed in the present work only models Markovian information transmission and does not inform us about other complementary brain communication mechanisms. Moreover, we used an undirected measure of information exchange between regions pairs, the mutual information, which is well adapted to the spatial and temporal resolution of fMRI recordings. Therefore, our framework cannot resolve 'star' relay information motifs with a central node being the source of information in the communication process, which is one of its limitations. Different measures of directed information exchange have been proposed in literature[62,63] and should be explored in future work in relation to the structural connectome architecture and communication mechanisms. It will be particularly interesting to test directed and more complex communication models on data with rich spatio-temporal information, such as intracranial EEG (see[19] for an example) and calcium imaging. By no means the proposed model aims at explaining the entire spectrum of communication mechanisms in brain networks. As such, absence of relay communication (i.e., violation of the data processing inequality) may indicate absence of any communication between those particular brain regions; communication limited to one single, direct structural connection (no parallel multi-step pathways); or communication through more complex information encoding mechanisms. Notwithstanding the evidence that selective and parallel Markovian pathways serve as important information streams for multimodal integration between unimodal and transmodal systems[37,38], we speculate that low parallel communication scores between transmodal regions may indicate predominance of more complex communication regimes in these areas. In addition, sensory input decoding within the highly clustered unimodal systems (diagonal entries of the parallel communication matrices, Fig. 3b) may be supported by synergistic processes within dense structural motifs[64]. How these macroscale communication mechanisms may have adapted to changing environments over the evolution of mammalian brains remains an exciting open field of research, to which the present work adds a perspective.

## Methods

### Human data

We used Magnetic Resonance Imaging (MRI) data of the Human Connectome Project (HCP), U100 dataset (HCP900 data release), which includes 100 unrelated healthy adults ("h-HCP" dataset, 36 females; mean age = 29.1 ± 3.7 years)[65]. All experiments were reviewed and approved by the local institutional ethical committee (Swiss Ethics Committee on research involving humans). Informed consent forms, including consent to share de-identified data, were collected for all subjects (within the HCP) and approved by the Washington University Institutional Review Board. All methods were carried out in accordance with relevant guidelines and regulations. MRI scans were performed on a 3 T Siemens Prisma scanner and included the following sequences: Structural MRI: 3D Magnetization Prepared Rapid Acquisition with Gradient Echoes (MPRAGE) T1-weighted, TR = 2400 ms, TE = 2.14 ms, TI = 1000 ms, flip angle = 8°, FOV = 224 × 224, voxel size = 0.7 mm isotropic. Diffusion-weighted MRI: spin-echo Echo-Planar Imaging (EPI), TR = 5520 ms, TE = 89.5 ms, flip angle = 78°, FOV = 208 × 180, 3 shells of $b$ value = 1000, 2000, 3000 s/mm$^2$ with 90 directions plus 6 $b$ value = 0 s/mm$^2$ acquisitions. One session of 15 min resting-state functional MRI (fMRI): gradient-echo EPI, TR = 720 ms, TE = 33.1 ms, flip angle = 52°, FOV = 208 × 180, voxel size = 2 mm isotropic, recorded with two phase-encoding directions (right-left and left-right). HCP minimally preprocessed data[66] were used for all acquisitions.

**Group-level structural connectivity.** A group-representative structural connectome between 100 cortical regions of interest (Schafer parcellation[67]) was obtained from the 100 unrelated HCP subjects. Different cortical parcellation resolutions were explored in supplementary analyses (200- and 400-region Schaefer parcellations[67]).

Briefly, diffusion-weighted scans were analyzed using MRtrix3[68], including the following steps: multi-shell multi-tissue response function estimation; constrained spherical deconvolution; tractogram generation with $10^7$ output streamlines. The Schaefer cortical atlas was used to parcellate the cortex into 100 (200, 400) regions and generate individual structural connectomes, from which a group-representative structural connectome was computed. The binary architecture of the group-representative connectome was obtained by including only the structural connections retrieved in 100% of the subjects. This step is meant to minimize the number of false positives in the group-representative network, and different consistency thresholds were investigated (100% to 50%; Supplementary Fig. 12). The group-representative connectome was then weighted by the Euclidean distance (in millimeters) between region pairs' centroids (Supplementary Fig. 1a). This choice was motivated by the exigence of homogenizing structural connections' weights across species (see also Mapping relay information-related pathways in brain networks).

**Individual functional information.** Resting-state fMRI data were pre-processed according to a state-of-the-art pipeline[30] including: general linear model regression of nuisance signals (removal of linear and quadratic trends; removal of motion regressors and their first derivatives; removal of white matter and cerebrospinal fluid signals and their first derivatives). 100 (200, 400) regional time series were obtained by averaging voxel-wise time series across all voxels belonging to each region of interest. The mutual information between region pairs was computed from the histograms of the z-scored time series, binned with a step of 0.5. This bin size was chosen by comparing real and null mutual information values, with null values obtained from multivariate gaussian data, and by assessing the fingerprinting accuracy[39] of mutual information across bin sizes (Supplementary Fig. 21). Only the first 800 time points (9.6 min) were considered for mutual information computation for consistency with other species data (Supplementary Table 1; other time series lengths were explored in supplementary analyses, Supplementary Fig. 13). Mutual information matrices obtained from left-right and right-left phase-encoding acquisitions were averaged to obtain a single $100 \times 100$ ($200 \times 200$, $400 \times 400$) mutual information matrix per subject (Supplementary Fig. 1c).

**Replication datasets.** Analyses were repeated considering sub-samples of the whole U100 dataset (Supplementary Fig. 15).

## Macaque data

We used structural and functional monkey data from 9 adult rhesus macaque monkeys (Macaca mulatta; 2 females) aged between 5 and 14 years scanned on a vertical Bruker 4.7 T primate dedicated scanner at Newcastle University[69] ("q-NCS" dataset). Raw data were publicly available through the Primate Data Exchange initiative[70] and included the following MRI sequences: Structural MRI: Modified Driven Equilibrium Fourier Transform (MDEFT) T1-weighted, TR = 2000 ms, TE = 3.75 ms, TI = 750 ms, voxel size = $0.6 \times 0.6 \times 0.62$ mm³. Two runs of 6.5 min resting-state fMRI: TR = 2600 ms, TE = 17 ms, voxel size = 1.2 mm isotropic. All animals were scanned awake. MRI data preprocessing included: T1-weighted volumes denoising[71], skull-stripping (FSL[72]), N4 bias field correction, spatial normalization to the F99 template obtained from the SumDB database (http://brainvis.wustl.edu/sumsdb/public_archive_index.html), and registration to fMRI native space (ANTs[73]); fMRI volumes were coregistered (FSL[74]), corrected for nuisance signals including six motion signals, average white matter and cerebrospinal fluid signals, and band-pass filtered to the band 0.01–0.15 Hz. Z-scored regional time series (Regional Map parcellation) of the two concatenated fMRI runs were used to compute individual mutual information values (bin size = 0.5). The fMRI scans were concatenated to reach a number of time points comparable with the other datasets (500 time points, 13 min).

**Group-level structural connectivity.** We used the whole-brain macaque structural connectome provided by TheVirtualBrain[75], which summarizes the brain connectivity between 82 regions of interest (Regional Map parcellation of Kötter and Wanke[76]) and includes inter-hemispheric connections. Briefly, the structural connectome was obtained by optimizing tractography-derived structural connectivity matrices with respect to a reference tracer-derived connectivity matrix and averaging across animals[75]. For cross-species consistency reasons, we considered undirected structural connectivity information. That is, in the final structural connectome, two regions are connected if at least one unidirectional connection exists between the two regions. Structural connections were weighted by the Euclidean distance (in millimeters) between region pairs' centroids (Supplementary Fig. 1a).

**Individual functional information.** Resting-state fMRI data were pre-processed by others, as previously described[75]. Briefly, the processing pipeline included motion correction, high-pass filtering, regression of white matter and cerebrospinal fluid signals, spatial normalization and smoothing. Z-scored regional time series (Regional Map parcellation) including 600 time points (10 min) were used to compute individual mutual information matrices (bin size = 0.5, consistently with other species) (Supplementary Fig. 1c).

**Replication dataset.** Analyses were repeated on an independent dataset from TheVirtualBrain project[75,77]. The fMRI dataset included 9 adult male rhesus macaque monkeys (8 Macaca mulatta, 1 Macaca fascicularis) aged between 4 and 8 years ("q-TVB" dataset). All methods were carried out in accordance with relevant guidelines and regulations and have been previously described[75]. Briefly, animals were lightly anesthetized before their scanning session and anesthesia was maintained using 1-1.5% isoflurane. The scanning was performed on a 7 T Siemens MAGNETOM head scanner included: Structural MRI: 3D MPRAGE T1-weighted sequence, 128 slices, voxel size = 0.5 mm isotropic. Diffusion-weighted MRI: EPI sequence, 24 slices, $b$ value = 1000 s/mm², 64 directions, recorded with two opposite phase-encoding directions. One session of 10 min resting-state functional MRI (fMRI): 2D multiband EPI sequence, TR = 1000 ms, 42 slices, $1 \times 1 \times 1.1$ mm³ voxel size.

## Mouse data

We used fMRI data of 10 C57Bl6/J adult male mice ("m-GG" dataset, <6 months old) subject to surgery for headposts placement, MRI habituation and awake fMRI acquisition, as previously described[29]. MRI acquisitions were performed at the IIT laboratory in Rovereto (Italy) on a Bruker BioSpin 7 T scanner and included a 32-min resting-state fMRI recording: single-shot EPI sequence, TR = 1000 ms, TE = 15 ms, flip angle = 60°, voxels size = $0.23 \times 0.23 \times 0.6$ mm³. fMRI preprocessing included exclusion of the first 2 min of recording, time series despiking, motion correction, nuisance signals regression (average cerebrospinal fluid and motion signals plus their temporal derivative and corresponding squared regressors), data censoring (Framewise Displacement >0.075 mm), band-pass filtering (0.01–0.1 Hz), spatial smoothing (FWHM = 0.5 mm) and spatial normalization[28]. Average time series were computed for 66 regions of interest, which represents a subset of the 78 Allen Brain Atlas regions (data for bilateral regions CA1, CA2, CA3, dorsal and ventral piriform nucleus, and frontal pole were not available). The first 600 time points (10 min) were used for the computation of individual mutual information matrices (z-scored time series binning = 0.5).

**Group-level structural connectivity.** A mouse structural connectome between 78 cortical regions covering the isocortex, cortical subplate, and hippocampal formation, as defined in the Allen Brain Atlas, was derived from published viral tracing data[78]. In more details, the binary

architecture of the structural connectome was assessed according to the following steps: (i) we considered the right-hemisphere ipsilateral and contralateral connections reported by ref. [78]; (ii) we symmetrized the right-hemisphere ipsilateral connections (i.e., we considered a connection between ipsilateral regions $i$ and $j$ to be present if at least one of the two tracts $(i \rightarrow j)$, $(j \rightarrow i)$ was detected); (iii) we duplicated the symmetrized ipsilateral connections to the left hemisphere (in absence of more detailed information, we therefore assume equal intra-hemispheric connectivity in the right and left hemispheres); (iv) we transposed the contralateral connections of the right hemisphere to the left hemisphere; (v) to minimize false positives due to minor tissue segmentation artifacts, we excluded connections with connectivity strength $<10^{-3.5}$, as suggested in[78], where the connectivity strength was defined as the total volume of segmented pixels in the target normalized by the injection site volume. The binary structural connectome was then weighted by the Euclidean distance between region pairs' centroids obtained from the Allen Brain Atlas (CCF v3, © 2004 Allen Institute for Brain Science. Allen Mouse Brain Atlas. Available from: http://www.brain-map.org/) (Supplementary Fig. 1).

**Individual functional information.** Resting-state fMRI data were pre-processed as previously described[79]. Briefly, the processing pipeline included motion correction, automatic brain masking, spatial smoothing (FWHM = 0.45 mm), high-pass filtering (0.01 Hz cut-off), and automated nuisance removal based on independent component analysis. Z-scored regional time series (78-region Allen Brain Atlas parcellation) including 600 time points (10 min) were used to compute individual mutual information matrices (bin size = 0.5) (Supplementary Fig. 1c).

**Replication datasets.** Analyses were repeated on two independent datasets. The first one included ten male wild-type mice aged 6 months ("m-AD3" dataset), available at https://openneuro.org/datasets/ds001890[80]. All methods were carried out in accordance with relevant guidelines and regulations and have been previously described[29,79]. Briefly, animals were anesthetized with 4% isoflurane before their scanning session and maintained with 0.5% isoflurane and a 0.05 mg/kg/h medetomidine infusion[55]. The scanning was performed on a 11.75 T Brucker BioSpin scanner and included: Structural MRI: spin-echo turboRARE sequence, TR = 2750 ms, TE = 30 ms, FOV = $17 \times 11$ mm$^2$, matrix dimension = $200 \times 100$ voxels, slice thickness = 0.35 mm. One session of 10 min resting-state functional MRI (fMRI): gradient-echo EPI sequence, TR = 1000 ms, TE = 15 ms, matrix dimension = $90 \times 60$ voxels. The second dataset included 51 male wild-type mice scanned at 3 months ("m-CSD1" dataset)[81]. MRI acquisitions were performed on a 9.4 T Brucker BioSpin system on anesthetized animals (3.5% isoflurane, maintained with 0.5% isoflurane and a 0.05 mg/kg/h medetomidine infusion) and included a 6-min resting-state fMRI recording: gradient-echo EPI sequence, TR = 1000 ms, TE = 9.2 ms, flip angle = 90°, field of view = $20 \times 17.5$ mm$^2$, matrix size = $90 \times 70$ voxels, slice thickness = 0.5 mm. FMRI volumes were preprocessed using the same pipeline as the m-AD3 dataset. The average time series of the 78 cortical regions (360 time points, 6 min) were z-scored and used to compute individual mutual information matrices (bin size = 0.5). Analyses were repeated considering sub-samples of the whole m-CSD1 dataset (Supplementary Fig. 15).

**Assignment of cortical regions to resting state networks**
For the human dataset, each cortical region was assigned to one the seven resting state networks (RSNs) defined by Yeo et al. and according to the Schaefer parcellation[67,82]. For the macaque dataset, each cortical region was first associated with one or multiple Brodmann areas according to the CoCoMac Regional Map of the macaque cortex[76,83–85]. Each Brodmann area was then assigned to one of the seven RSNs defined by Yeo and colleagues[82] using a majority voting procedure and published atlases in MNI space[86]. Finally, Regional Map

regions of the macaque cortex were assigned to Yeo RSNs with a similar majority voting procedure (Supplementary Fig. 3). For the mouse dataset, each cortical region was assigned to one out of 6 RSNs as identified by Zerbi and colleagues using independent component analysis of resting-state fMRI data[87]. The assignment was done through a majority voting procedure (Supplementary Fig. 4). Note that the default mode network (DMN) has been consistently identified in humans[88], macaques[89] and mice[90,91], suggesting a conservation of this network across mammalian species. In our mouse cortex subdivision[87], the DMN includes bilateral hippocampal regions (CA1, CA2, CA3 hippocampal fields, subiculum and dentate gyrus), and lateral (entorhinal and temporal association areas) and prefrontal (infralimbic, prelimbic and perirhinal areas) isocortices, while it excludes other regions which have been reported by others, such as the retrosplenial cortex[92]. For all species, RSNs were assigned to unimodal or transmodal systems according to established cortical subdivisions[38].

**Mapping relay information-related pathways in brain networks**
In this work we introduce an approach to probe relay information-related pathways from multimodal neuroimaging data. The approach builds upon and extends an information theoretical framework proposed in previous work[32] and aims at identifying polysynaptic (multi-step) structural paths compatible with relay information transmission in macroscale brain networks. Information theory is a branch of mathematics that studies the transmission of information through communication systems[33] and has found several applications in neuroscience[93,94]. It allows the analysis of noisy data, such as the fMRI ones.

**Structural brain network and structural paths.** Let's consider a structural brain network as an undirected graph $G \equiv \{V, W\}$ formed by a set of $N$ nodes $V = \{v_1, v_2, ..., v_N\}$ and a connectivity matrix $W = [w_{i,j}]$, with $w_{i,j} > 0$ distance between directly connected region pairs $v_i, v_j$ and $w_{i,j} = \infty$ otherwise. In this work we assigned $w_{i,j}$ equal to the Euclidean distance (in millimeters) between the centroids of regions $v_i, v_j$. This choice has two motivations. First, the distance between region centroids can be easily computed across different datasets, thus allowing to select homogeneous structural connectivity weights across species. Second, this choice conceptually links information transmission in brain networks with the sender-channel-receiver schematics proposed in electronic communication by Shannon[32,95]. A path between a source node $v_i$ and a target node $v_j$ is a sequence of pairwise connected and non-repeating nodes $\Omega_{i,j} = \{v_i, v_a, v_b, ..., v_j\}$. The shortest path $\Omega_{i,j}^{SP}$ between regions $v_i, v_j$ is the path of minimal length (i.e., minimal Euclidean distance, in the case of this work) connecting the two regions. The path length is computed as the sum of edge weights along the path. In this work we identified the first $k = 5$ k-shortest paths $\Omega_{i,j}^{k-SP}$ connecting each region pair $v_i, v_j$[96]. $K$-shortest path ensembles identify meaningful trade-offs between efficiency and resiliency for putative communication processes in brain networks[18]. The choice of $k$ was dictated by the fact that, for $k = 5$, all edges of the structural brain network participate in at least one k-shortest path[18]. In fact, previous results showed that, when investigating the fraction of network connections participating in at least one multi-step path as a function of k, this fraction increases with $k$ and rapidly converges to a plateau at $k \sim 5$[18]. This finding was confirmed in our study where, when considering the first 5-shortest paths between any pair of brain regions, we found that 91%/87%/94% of network connections were used in at least one multi-step path for humans/macaques/mice, respectively.

**Functional information along structural paths.** Each node $v_i$ is associated with a neural activity-related fMRI time series $X^i$ that can be interpreted as the realization of a discrete random variable with probability mass function $p_i(x^i)$. The amount of shared information between

two random variables can be quantified as their mutual information $I(X^i, X^j) = \sum_{x^i \in X^i} \sum_{x^j \in X^j} p_{i,j}(x^i, x^j) \log_2(p_{i,j}(x^i, x^j)/(p_i(x^i)p_j(x^j)))$, with $p_{i,j}(x^i, x^j)$ joint probability distribution between $X^i, X^j$. The sequence of pairwise mutual information values along a structural path $\Omega_{ij}$ with respect to the source node $v_i$ is defined as $\Phi_{ij} = \{I(X^i, X^a), I(X^i, X^b), \dots, I(X^i, X^j)\}$. We estimated the fMRI time series probability mass functions from the z-scored time series' histograms with appropriate binning. Different bin sizes between 0.05 and 2.00 were explored and evaluated with respect to (i) corresponding mutual information values for multivariate Gaussian processes $\aleph(0, I)$; (ii) individual identifiability scores[39]. We selected the smallest bin size for which (i) the mutual information values obtained from real data (h-HCP dataset) were larger than expected for a multivariate Gaussian process $\aleph(0, I)$, and (ii) the individual identifiability score reached a maximum plateau (Supplementary Fig. 22).

**Data Processing Inequality (DPI)**. The DPI, a fundamental principle of information theory, states that the amount of information available at a target node $v_j$ about a source node $v_i$ cannot be increased through operations performed along the transmission path. Mathematically, the DPI states that if $X^i - X^a - X^j$ is a Markov chain, then $I(X^i, X^a) \geq I(X^i, X^j), I(X^a, X^j) \geq I(X^i, X^j)$, i.e., the mutual information does not increase along the chain[33]. Note that the double inequality condition derives from the fact that a Markov chain has no directionality information, i.e., if $X^i - X^a - X^j$ is a Markov chain, then $X^j - X^a - X^i$ is also a Markov chain. The DPI can be extended to Markov chains of any length. Conceptually, the DPI embeds two assumptions about the information transmission process: the first one is that (neural) messages transmitted through the structural infrastructure (brain network) can keep at most the same amount of information present at the source region (information decay). The second one is that (neural) messages do not contain memory of the transmission process itself and communication happens in a Markovian fashion (memoryless transmission).

**Identification of relay information-related pathways in brain networks**. We used the DPI to test (deviation from) Markovian behavior. Each k-shortest structural path was labeled a relay information-related pathway if the DPIs along the paths were satisfied. Note that here we use the wording relay pathways in Shannon's sense. That is, we aim to characterize the presence of memoryless information transmission processes, with information decay along the path measured through mutual information values. Note that the DPI framework concerns multi-step paths only, since the data processing inequality cannot be assessed on 1-step connections. Therefore, our results concern multi-step paths only, which is in line with the concept of parallel communication. Note also that the considered structural networks, which were built from diffusion MRI and track tracing data, do not include multiple "parallel" 1-step connections. Each pair of brain regions can be connected by at most one 1-step connection.

**Parallel communication scores (PCSs)**. We define the parallel communication score $PCS_{i,j}^n$ between a pair of brain regions $v_i$, $v_j$ as the number of k-shortest paths connecting the two regions which respect the DPI, with $n$ indicating the subject. Note that, given the choice of $k = 5$, PCS scores can assume integer values between 0 and 5, and that $PCS_{i,j}^n = PCS_{j,i}^n$. A PCS score equal to 0 is interpreted as absence of (Markovian) information transmission between two regions; a PCS score equal to 1 is interpreted as presence of selective information transmission through a single information-related pathway; PCS scores larger than 1 are interpreted as presence of progressively increasing parallel information transmission, with transmission through multiple parallel pathways (Fig. 1). PCS scores were computed for every pair of brain regions and every subject, for all investigated datasets. Parallel communication information was summarized at the group-level by

computing a group-average parallel communication matrix $PCS^{avg}$ for each dataset, and its corresponding histogram (Fig. 2). In addition, node-average, RSN-average, and system-average (unimodal, transmodal, cross-modal) PCS scores were computed by averaging the parallel communication scores over the corresponding region pairs (Fig. 3).

**Null models**. Three null models were defined in this work. A first null model was defined by randomly shuffling the raw fMRI time series across brain regions while preserving the original structural connectivity information (Supplementary Fig. 6). Note that with this randomization we are preserving the statistical properties of both the original functional and structural data, since we are merely rearranging spatially fMRI time series across the brain network. Parallel communication matrices were then computed for each randomization following the above-described procedure. For each dataset, the randomization was repeated 3000 times per subject, which allowed to build 3000 group-average parallel communication matrices (Supplementary Fig. 7). Each region pair was therefore associated with a null distribution of group-average PCS values including 3000 elements. To assess whether group-average PCS scores observed in real data could be trivially explained by the structural connectivity architecture and the multivariate statistical properties of fMRI data, which are both preserved in the null model, we adopted two strategies. The first one consisted of PCS scores screening by z-scoring individual group-average scores $PCS_{i,j}^{avg}$ with respect to the corresponding null distribution; z-scored where thresholded at 1.96 (Supplementary Fig. 8). The second strategy consisted of analyzing the PCS scores with false discovery rate (FDR)-corrected $p$ values < 0.05 (FDR <0.05), with $p$ values computed as the number of entries in the null distribution exceeding the real PCS score (Supplementary Fig. 9).

A second null model was defined by populating the network nodes with iid Gaussian noise with mean 0, variance 1, and the same number of time points as in experimental data (100 simulations). This scenario corresponds to the case of absent communication (Supplementary Fig. 19).

A third null model was developed for mutual information (MI)-based functional connectomes, which preserves their spatial autocorrelation with respect to the underlying distance-weighted structural connectivity matrix. Specifically, similarly to the BrainSMASH algorithm[41], we first quantified the level of spatial autocorrelation in the original individual MI matrices with a variogram. This is the target variogram for the output null MI matrices. Then, the original individual MI matrix is randomly permuted. Spatial autocorrelation among the samples is reintroduced by smoothing the permuted map with a distance-dependent kernel (similarly to BrainSMASH, here we employed an exponentially decaying gaussian kernel). Afterwards, the smoothed MI matrix's variogram is computed and then regressed onto the variogram for the original MI matrix. The goodness-of-fit is quantified by computing the sum of squared error (SSE) in the null vs. original variogram fit. Finally, for every individual, we picked the null MI matrix with the best goodness-of-fit over 100 permutation runs (Supplementary Fig. 20).

**Computation of fMRI time series' signal-to-noise ratio (SNR)**
From a technical viewpoint, scanners and protocols are significantly different between species as they have been optimized for their particular usage. It is therefore a reasonable concern that the ultimate quality of the fMRI data in terms of SNR might be confounded with the measure of interest, the PCS. To tackle this issue, we investigated possible correlations between nodal SNR and average PCS scores at the individual level (Supplementary Fig. 11). Our brain-communication method takes preprocessed, regionally averaged, and normalized (z-scored) fMRI time series as an input. The nodal SNR was therefore quantified as the ratio of the time series' low- to high-frequency power contents, where the frequency cut-off is taken as half of the Nyquist frequency. The rationale is that low-

frequency content can be assumed to be dominated by hemodynamic contributions, and high-frequency content by noise components, respectively. As shown in Supplementary Fig. 11a, the SNR of human (h-HCP) and macaque (q-NSC) data is comparable, being slightly higher for macaque. The SNR for mouse data (m-GG) is lower, which can be explained by the faster hemodynamic signals in this species and thus less power to be expected in the low-frequency part. Supplementary Fig. 11b shows that in none of the species a relationship exists between the SNR of the fMRI time series and the PCSs across brain regions. Therefore, both inter- and intra-species measures of SNR and their lack of relation with PCS indicate that basic scanning protocol differences can be excluded as a confounding factor of our results.

## Subject identifiability analysis

For each investigated non-anesthetized dataset, fMRI time series were split into two parts of equal duration and considered as test and retest data. From these, test and retest parallel communication matrices were computed for each subject. An identifiability matrix summarizing test–retest subjects' similarities was then obtained for each dataset. Diagonal entries of the identifiability matrix represent subjects' self-similarity between test and retest data ("Iself"); outside-diagonal entries represent inter-subject similarity ("Iothers") (Fig. 4)[39]. The similarity between test and retest parallel communication matrices was assessed with the Jaccard index, defined as the size of the intersection divided by the size of the union of two label sets. For example, a Jaccard index equal to 0.3 indicates that 30% of brain region pairs have exactly the same PCS score, which can take integer values between 0 and 5. The level of individual identifiability was quantified with the success rate (SR) defined as the percentage of test subjects whose identity was correctly predicted out of the total set of retest subjects[40]. The subject identifiability analysis was repeated when considering only region pairs with, on average, low (high) PCS scores for the computation of test-retest similarities. Different thresholds defining low (high) PCS scores were explored (Supplementary Table 3).

## Reporting summary

Further information on research design is available in the Nature Portfolio Reporting Summary linked to this article.

## Data availability

All data used in this study are available through open-source repositories. The human h-HCP dataset[65] is available at https://db.humanconnectome.org. The macaque q-NCS dataset[69] is available through the Primate Data Exchange (PRIME-DE) initiative[70]. The macaque q-TVB dataset[75] is available at OpenNEURO[77]. The mouse m-GG dataset[29] is available at https://data.mendeley.com/datasets/np2fx99hn6/2. The mouse m-AD3 dataset is available at OpenNEURO[80]. The mouse m-CSD1[81] dataset is available at the XNAT Data Repository https://central.xnat.org/ (Project_ID: CSD_MRI_MOUSE). A sample dataset generated in this study from open-source raw and processed data, including brain k-shortest paths and mutual information matrices of the three species, is available at A.Gr.'s GitHub repository (https://github.com/agriffa/BrainComm_mammalian_evolution). Source data are provided with this paper.

## Code availability

The code and sample brain data to reproduce the main results of this study are available at A.Gr.'s GitHub repository (https://github.com/agriffa/BrainComm_mammalian_evolution).

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

## Acknowledgements

We thank Prof. Joaquín Goñi, Prof. Dante Mantini, Prof. Martijn Van den Heuvel and Dr. Kelly Shen for their critical suggestions and helpful discussions. We thank Dr. Maria Giulia Preti and Dr. Raphaël Liégeois for the processing of human data. We thank sci-draw.io for the brain schematic. Human data were provided by the Human Connectome Project, WU-Minn Consortium (Principal Investigators: David Van Essen and Kamil Ugurbil; 1U54MH091657) funded by the 16 NIH Institutes and Centers that support the NIH Blueprint for Neuroscience Research; and by the McDonnell Center for Systems Neuroscience at Washington University. A.Gr. acknowledges funding from the Ernest Boninchi Foundation project "BrainCom—Communication dynamics in system-level brain networks: Novel methodology with application to reversible dementia" and the Swiss National Science Foundation (Grant No. 320030_173153). E.A. acknowledges financial support from the SNSF Ambizione project "Fingerprinting the brain: network science to extract features of cognition, behavior and dysfunction" (Grant No. PZOOP2_185716). A.Go. acknowledges funding from the European Research Council (ERC, #DISCONN; no. 802371), the Brain and Behavior Foundation (NARSAD, 25861), the NIH (1R21MH116473-01A1) and the Telethon foundation (GGP19177).

## Author contributions

A.Gr., E.A., D.V.D.V conceptualized the study and developed the methodology. A.Gr., E.A., M.M., J.D. developed the code. A.Gr., M.M., J.D. analyzed the data. A.Gr. prepared the figures. D.G.B., A.Go., J.G., D.V.D.V., G.A., E.A., A.Gr. contributed to the resources. A.Gr., E.A. wrote the original draft. All the authors contributed to the review and editing of the draft.

## Competing interests

The authors declare no competing interests.
