## [Peer Review File · Nature Communications]

Evidence for increased parallel information transmission in human brain networks compared to macaques and miceReviewer #1 (Remarks to the Author):

Griffa and colleagues present a manuscript that investigates the differences in information transmission across species by performing network-based analyses on multimodal (fMRI and sMRI). Inspired by information theory, they apply the Data Processing Inequality (DPI) on brain network data, from which they derive the Parallel Communication Score (PCS). This analysis hinges on the assumption that the brain transmits information under a Markovian framework. Within this framework, the authors conclude that the number of structural paths that obey the DPI reflect the number of parallel information transmission pathways. The main findings are that across mammalian species (mouse, monkey, human), PCS scores are organized according to their evolutionary complexity (humans, monkeys, then rodents). Humans also have the greatest PCS across unimodal and transmodal cortices (i.e., cross-modal PCS). Finally, the authors show that PCS network organization is unique across individuals.

Overall there appears to be an ambitious amount of work that has gone into this study, from the processing and analysis of multiple species and multiple data modalities. However, I have a number of major concerns regarding the validity of the analyses, interpretations, and conclusions. First, while the analysis is interesting and creative, it is highly speculative in that it assumes that the brain transmits information under a Markovian framework. (The paper does not rule out if the brain uses any non-Markovian framework.) Second, the comparison across species (as presented in the main text), is highly confounded by the fact that both monkeys and rodents were not anesthetized. The supplementary figures that show data for unanesthetized monkeys/rodents are not entirely convincing, as it shows a strong anesthetization effect, while also showing that unanesthetized rodents have an overall higher PCS than unanesthetized monkeys, which is contrary to the authors' main claims (Supp. Fig 14). Finally, while I acknowledge the difficulties (and ambition) of trying to analyze datasets from many different sources, the differences in MRI scanning protocols (e.g., TR length) give many degrees of freedom that were not fully addressed. I detail these concerns (and others) below.

Major

* A major concern is that the results do not clearly reflect the hypothesized differences of PCS across mammalian species when controlling for anesthesia. Specifically, unanesthetized mice had greater PCS than unanesthetized monkeys (Supp Fig. 14d vs f). This appears to be inconsistent with the claims made in the Results section that reference the supplementary analysis (line 400). This appears to be a misleading mischaracterization of supplementary data. In general, I think the unanesthetized results should all be presented in the main text, and the anesthetized results placed in the supplementary (to avoid any confusion).

* I'm not an expert in evolutionary biology, but is it common to phrase the transition from mouse to monkeys to humans as "evolutionary complexity"? From my naive knowledge on evolution, all three species are highly specialized for their current (contemporary) environments, and I'm curious to know if it's common to think of humans as a "more evolved form" of a mouse, for example. This seems to be an oversimplification. Otherwise, citations are warranted.

* The choice of 5 shortest structural paths seems somewhat arbitrary to me. If I understand correctly, Supplementary Fig. 1b seems to indicate that a majority of the shortest paths are only 1-step connections. It follows that most likely, the 5 shortest connections will likely be 1-step connections. What is the significance of DPI of a 1-step connection? Can it truly be considered 'parallel' if this analysis is reduced to if the top 5 1-step connections obey the DPI (whereas the schematic in Fig. 1 appears to be an illustration for 2-step connections)?

* Another question I had about the SC matrix was that it appears that SC matrices were binarized, and then the weights were modified by their Euclidean distances. However, wouldn't this bias the shortest paths to be very close range connections? Would any long range connections be included? Perhaps it could be helpful to provide some examples of what long range connections are included for a few example regions.

* Evaluating and comparing relay communication patterns that are unique to individuals. How

were the fMRI recordings split? Were independent imaging sessions tested apart? Or were they data points within the same session? It wasn't immediately clear from the Methods, but for statistical independence sessions/data should be assessed from independent scans. Otherwise individual fingerprinting would be confounded if data were taken from the same scan session.

* I would encourage the authors to be a bit more careful with terminology. Throughout the manuscript, they often equate PCS with the brain's information communication strategy (e.g., line 420 (Discussion)). However, the measure of PCS is a information communication strategy that is imposed by the author (experimenter-derived measure), rather than a strategy that is actually implemented by the brains. The authors should clarify this in the Introduction/Discussion.

* The measurement of SNR seems to be confounded by the differences in TR. Since (according to the methods) SNR is just the ratio of low v. high freq power contents, determined by the half of the Nyquist frequency, then SNR measurements across scans with different TR lengths are likely incomparable.

Minor

* The mismatch between using group-level SC matrices versus individual-level FC (or mutual information) analyses seems problematic. I appreciate the difficulties in estimating individual SCs due to high variance of diffusion scans, but this seems like it should be addressed as a limitation. Moreover, this might suggest that the individual fingerprinting of PCS (Fig. 4) derives entirely from the mutual information analysis, rather than the combination of mutual information + SC + DPI analysis. Does fingerprinting of PCS provide something over and above just functional fingerprinting (using, for example, mutual information)?

* How would PCS be assessed if using just linear correlation rather than mutual information? What is the rationale for using mutual information, given that correlations are just the linear equivalent of MI?

* I wonder if species-specific PCS differences across unimodal/transmodal areas could be in part explained by differences in the structure-function tethering across those species? The specific work that comes to mind is: (Baum et al., 2020; Vázquez-Rodríguez et al., 2019).

* Is merely shuffling the mutual information matrix appropriate as a permutation/null test (e.g., in Fig. 2)? Since both fMRI and the SC matrix are both confounded by spatial autocorrelation, my sense is that arbitrarily shuffling the mutual information matrix without preserving any spatial autocorrelation may not be the appropriate null to test whether there are true differences in PCS across unimodal-transmodal areas.

* In supplementary Figure 17, what do the p-values in the top-left corner of each graph reflect? Same for Figure 3 and related figures. The specific statistical test is reported, but not what is being compared.

References

Baum, G.L., Cui, Z., Roalf, D.R., Ciric, R., Betzel, R.F., Larsen, B., Cieslak, M., Cook, P.A., Xia, C.H., Moore, T.M., Ruparel, K., Oathes, D.J., Alexander-Bloch, A.F., Shinohara, R.T., Raznahan, A., Gur, R.E., Gur, R.C., Bassett, D.S., Satterthwaite, T.D., 2020. Development of structure–function coupling in human brain networks during youth. *Proc. Natl. Acad. Sci.* 117, 771–778. <https://doi.org/10.1073/pnas.1912034117>

Vázquez-Rodríguez, B., Suárez, L.E., Markello, R.D., Shafiei, G., Paquola, C., Hagmann, P., Heuvel, M.P. van den, Bernhardt, B.C., Spreng, R.N., Misic, B., 2019. Gradients of structure–function tethering across neocortex. *Proc. Natl. Acad. Sci.* 116, 21219–21227. <https://doi.org/10.1073/pnas.1903403116>

Reviewer #2 (Remarks to the Author):

The manuscript „The evolution of information transmission in mammalian brain networks“ by Griffa et al describes a novel methodology to quantify functional connectivity between brain regions based on the number of parallel structural connections that are used for information transmission. They use an information theoretic approach assuming that mutual information between a signal observed in a source and along the pathways should be decreasing monotonically, and that the pathway itself is memoryless.

First, the structural connectome among brain regions is derived based on diffusion tensor imaging data. Then, connections between brain regions are evaluated based on paths with a defined length through the structural network.

Overall this approach adds a new perspective to the assessment of functional networks. Even though the authors suggest several insights based on the comparison of mouse, macaque, and human brain, the primary contribution is methodological.

In the following a few points:

It would facilitate reading, if the paper would start with describing the claim of counting parallel structural connections that are used for signal transmission as observed in functional magnetic resonance imaging (fMRI). After that, introduce the information theoretic approach used to evaluate this. Currently for a long initial stretch, the only point readers learn about the novelty is that it is an „information theoretic approach“.

Provide a motivation what exact insight this adds to our understanding of brain architecture and its evolution. What are many parallel connections associated with? Are there prior findings, or hypotheses regarding the functional brain architecture that explain the rationale? What are the „different communication strategies“ associated with parallel paths used at the same time. Please explain, and clarify this - as one of the main claims is regarding evolution, this is an important part. Maybe one can expand figure 1 to explain this overall concept.

Is it correct that the paths run along the cortex? That is, they are several hops from cortical region to cortical region? I assume this is the case, since your results are based on fMRI, but it is worth clarifying early.

Figure 2 is overall helpful, but it would be better, if you provide a visualization of the brain regions corresponding to the block-diagonal segments.

Please clarify the term „relay communication density“. Is there some context that can help understand its significance in this paper?

If I understand correctly, then the conclusion of Figure 2 is one of the main conclusions of the paper: communication becomes more parallel when comparing mouse to macaque to human. Is it possible that this is in part an artefact of increased brain size, corresponding increased relative DTI resolution, and therefore overall higher numbers of structural connections? The authors perform analysis to create a null-connectivity model by randomly permuting the assignment of time courses to cortical regions - that makes sense. The number of parallel connections remains significant after this (line 371). However, this statement does not extend to the differences in parallelism between the three species. The results in Figure 7 suggest that after this z-score normalization of parallel transmission pathways, the 3 species are almost the same. Please clarify: is there still a significant gradient of increasing parallelism, when controlling for the number of connections? This is a central claim of the paper, and needs proper testing.

To better understand the meaning of the parallel transmission pathway characteristic, please compare with other comparable approaches to connectivity, such as standard functional connectivity measures, number of structural connections (i.e., pretending that all structural connections are used for transmission). An interesting question is also how this differs for different brain areas as shown in Figure 3. Are there possible relationships to current approaches of

assessing functional brain architecture and evolution?

Several minor points:

line 153: „at most“ is not an assumption but a necessity if I understand correctly. Please clarify or formulate precisely.

line 222: similar for 3 species but decaying right after ... what does that mean?

line 371ff the parallelism or also it's increase remained significant when compared to null hypothesis?

is this independent from overall connectivity of a region? compare to count of connections from a region of all paths are counted as valid path. or is this equivalent to the time series shuffling please clarify

please check grammar, e.g., line 58: give -> gives

line 62: what do you mean by „virtually subtend all aspects“? Please make clear statements. Do you mean „subtend almost all aspects“

Reviewer #3 (Remarks to the Author):

This paper aims to explore the information transmission mechanisms in mammalian brain networks. First, the information transmission in brain networks is modeled as (multi-hop) relay communications. Second, the data processing inequality (DPI) in information theory is used to find all the structural paths between two brain regions which can serve as a relay communication channel. More specifically, a structural path is labeled as a communication channel if the pairwise mutual information values do not increase along the unidirectional path; otherwise, it is not a communication channel. For example, consider a path $A \rightarrow B \rightarrow C \rightarrow D$ and let $I(X,Y)$ denote the mutual information between regions X and Y . If $I(A,B) \geq I(A,C) \geq I(A,D)$, then $A \rightarrow B \rightarrow C \rightarrow D$ is labeled as a communication channel, otherwise it is not a communication channel.

Based on this approach, the authors analyzed the brain networks for mice, macaques, and humans, and found that, on average, the number of parallel communication path progressively increase from mice to macaques to humans, and the relay communication/routing pattern varies across species and different individuals in the same species. These conclusions are consistent with our common sense: parallel communications are critical for advanced cognitive functions in the human brain, and different people may have different connectivity patterns in the brain network due to different experiences and trainings. The paper is well written and easy to follow. This reviewer has the following comments:

1. A path that follows DPI may not be a communication channel.

First, let's assume that the relay model is correct. While it is true that a memoryless relay communication channel generally follows DPI, a path that follows DPI may not be a communication channel.

Let's look at an example. Before that we need to understand what a communication channel is. I believe that we all have the experience that, sometimes, our phone call is dropped because the signal fades, or our Zoom link does not work when the network speed is too low. This means that a communication is possible only when the connectivity between the source and destination is strong enough to support effective information exchange between them. Consider a structural path $A \rightarrow B \rightarrow C \rightarrow D$, even if there is only noise in the observed signal, we are likely to observe that $I(A,B) \geq I(A,C) \geq I(A,D)$ (with all the mutual information values been nonzero as the noise there may not be independent), simply because that distance between brain regions increases along the path. However, in this case, the path follows DPI, but there is actually no information transmission going on.

Second, since mutual information reflects the mutual dependence between the brain regions and is nondirectional, a path that follows DPI may not be a communication channel.

Consider an example $A \leftarrow B \rightarrow C$ (just as in the case that B is a friend for both A and C, but A and C do not really know each other), we are likely to have $I(A,B) \geq I(A,C)$, and then the proposed approach will assume that the $A \rightarrow B \rightarrow C$ is a communication channel but this is not true.

2. A path that does not follow DPI may be a communication channel.

Let's look at an example. Consider a path $X \rightarrow Y \rightarrow Z$. Assuming X and Z are students, and Y is a teacher. X reports to Y about what he is doing; Y then asks Z to contact X and does the same thing X is doing; Z contacts X and they exchange information. Recall that physical meaning of mutual information is the amount of information successfully exchanged between the two regions. In this example, we are going to see that $I(X,Z) \geq I(X,Y)$, it does not follow DPI; however, $X \rightarrow Y \rightarrow Z$ surely is a communication path.

Coming back to the brain, if each brain region just performs information relay without, then the communication will not be meaningful and it will be impossible for us to perform any cognitive thinking. The brain might be more likely to follow a receiving-processing-transmitting model rather than a simple relay communication model.

3. Unidirectional information transmission, in general, is measured using directed information rather than mutual information. Mutual information and directed information are equal only when there is no feedback in the channel. For directed information, the authors may want to refer to the following paper:

J. L. Massey and P. C. Massey, "Conservation of mutual and directed information," Proceedings. International Symposium on Information Theory, 2005. ISIT 2005., 2005, pp. 157-158, doi: 10.1109/ISIT.2005.1523313.

Overall, the topic is very interesting, and I appreciate the numerical analysis the authors have carried out. However, unfortunately, I am very sorry to say that using the DPI approach for communication path detection in brain networks is not technically sound.

I believe that analyzing brain network using information theory is perhaps the right direction, but this particular approach should be rethought.

Response to Reviewers

Reviewer #1 (Remarks to the Author):

Griffa and colleagues present a manuscript that investigates the differences in information transmission across species by performing network-based analyses on multimodal (fMRI and sMRI). Inspired by information theory, they apply the Data Processing Inequality (DPI) on brain network data, from which they derive the Parallel Communication Score (PCS). This analysis hinges on the assumption that the brain transmits information under a Markovian framework. Within this framework, the authors conclude that the number of structural paths that obey the DPI reflect the number of parallel information transmission pathways. The main findings are that across mammalian species (mouse, monkey, human), PCS scores are organized according to their evolutionary complexity (humans, monkeys, then rodents). Humans also have the greatest PCS across unimodal and transmodal cortices (i.e., cross-modal PCS). Finally, the authors show that PCS network organization is unique across individuals.

Overall there appears to be an ambitious amount of work that has gone into this study, from the processing and analysis of multiple species and multiple data modalities. However, I have a number of major concerns regarding the validity of the analyses, interpretations, and conclusions. First, while the analysis is interesting and creative, it is highly speculative in that it assumes that the brain transmits information under a Markovian framework. (The paper does not rule out if the brain uses any non-Markovian framework.)

We thank the Reviewer for the thorough evaluation of our work.

We agree with the Reviewer that the hypothesis of Markovian information transmission in the brain is speculative. However, we would like to underline that our Markovian framework is a high-level approximation of communication at the whole-brain system level, and is not meant to model communication mechanisms at the neuronal level. In line with other complex system models, we traded off biological detail for the ability to capture the emergence of global patterns compatible with the observed data. In the first place, we therefore use the Markovian framework (DPI) as a model to explain the observed BOLD signals at their temporal (seconds) and spatial (mm) resolution.

In a way, this approach can be considered a variant of more traditional functional connectivity (FC) analyses. FC does not explain how the brain works at the neuronal level, and yet it is an effective (linear) model for brain signals even though it does not capture all the subtleties inherent to large-scale brain signaling. In this respect, the DPI framework is a first step towards moving beyond the simplistic FC model, by applying well-grounded information theoretical principles that help bridging structure and function together and modeling information exchange between brain macro-regions.

Secondly, it is interesting to note that a majority of network models for brain communication do implicitly make the assumption of a Markovian system without explicitly probing it. Examples include information routing, flow-based and random-walk models, but also some epidemic

spreading and threshold models (Avena-Koenigsberger, Misic and Sporns, 2018). Our work contributes to this literature by explicitly assessing the plausibility of Markovian processes in large-scale brain networks through the DPI framework.

Finally, this work investigates relay (i.e., Markovian) information pathways only, but does not rule out the possibility of higher-order mechanisms such as (non-Markovian) causal communication or interferences - as recent works start to investigate (Luppi et al., 2022; Santoro et al., 2023). Indeed, it is likely that Markovian and higher-order mechanisms contribute to large-scale brain network communication. Nonetheless, the results presented here show that Markovian pathways are present in different mammalian brain networks, and could represent an effective model of information transmission in brain data. We are aware that future work should include and characterize also the aforementioned higher-order communication mechanisms. We have elaborated more on this aspect in the Discussion section, which now reads:

“...Intriguingly, brain communication models mostly rely on the assumption of memoryless (Markovian) information transmission⁵⁰. This hypothesis is pervasive in network neuroscience³⁷ but has never been formally probed in the brain. Our work adds to the field by introducing a new framework that explicitly models relay communication pathways from multimodal brain data, in a way that is grounded in fundamental information-theoretic principles. The framework models memoryless information transmission in brain networks by introducing an empirical way to assess deviations from Markovity through the data processing inequality³⁵. Our results show that Markovian communication is consistent with brain data of different mammalian species, is not limited to the shortest structural path, but involves multiple and less optimal structural paths in a way that is species-dependent and consistent with the phylogenetic level of the investigated species.

Polysynaptic memoryless information transmission is a simple model of higher-order communication. There is no reason to assume that macroscale neural communication is limited to such a particular form. Brain network hierarchies may confer neural signals a memory of the regions previously visited along a path, thus modifying neural communication pathways in a context-dependent manner⁵⁰. This process would result in non-Markovian communication regimes. The brain may also be modeled through complex multi-object interactions not attributable to information transmission alone, such as synergistic or modulatory behaviors between multiple brain regions; feedback loops; local transformation (non-linear processing) of information^{19,40,58,59}. Biologically, these more complex communication patterns may shape important features of the mammalian brain, such as cortical temporal hierarchies^{52,53} or receptive time windows for attentional processes⁵⁴, and are worthy to investigate in future work. The framework proposed in the present work only models Markovian information transmission and does not inform us about other complementary brain communication mechanisms.”

Second, the comparison across species (as presented in the main text), is highly confounded by the fact that both monkeys and rodents were not anesthetized. The supplementary figures that show data for unanesthetized monkeys/rodents are not entirely convincing, as it shows a strong

anesthetization effect, while also showing that unanesthetized rodents have an overall higher PCS than unanesthetized monkeys, which is contrary to the authors' main claims (Supp. Fig 14).

We thank the Reviewer for pointing out this important aspect, which brought us to perform additional statistical analyses and provide several clarifications.

We agree with the Reviewer that results may suggest an anesthetization effect for what concerns group-average parallel communication scores (PCSs) of macaques and mice (SI Fig. 14 - now SI Fig.15). For this reason, we performed new statistical analyses to assess differences of group-average and individual-level PCS scores between all datasets' pairs (i.e., all main and replication datasets, including both anesthetized and non-anesthetized animals). Moreover, to avoid any misinterpretation, we moved results on non-anesthetized animals from SI to the main text, and results on anesthetized animals from the main text to SI. We report below the new main Figures 2 and 3, which now include only data from non-anesthetized humans, macaques and mice, and the new SI Fig. 2 and 5, corresponding to the new statistical analyses.

At the group-average level and when considering all the six datasets (i.e., main and replication datasets, SI Fig. 2), the gap in parallel communication between humans and other mammalian species (macaques and mice) was strongly significant, while the differences between macaques and mice were minor. We observed that human group-average PCSs are different (larger) than macaques and mice PCSs, independently from anesthesia (average effect size = 0.28). We also found relatively minor differences in group-average PCSs between non-anesthetized and anesthetized mice only (not in macaques), with non-anesthetized mice showing larger average PCSs than anesthetized animals (m-GG vs m-AD3 and m-CSD1 murine datasets, average effect size = 0.13). Non-anesthetized mice also showed slightly larger group-average PCSs than anesthetized macaques (m-GG vs q-TVB dataset, effect size = -0.13), as the Reviewer pointed out.

Figure 2 Parallel communication gap across mammalian species. Left: drawing of human, macaque, and mouse brains; each row in the figure corresponds to one species. **(a)** Box plots representing the percentage of short paths in individual brain networks used for relayed communication (i.e., respecting the DPI). Each colored dot represents an individual; gray dots represent species-specific null distributions obtained from permutation of mutual information values (Online Methods); circles and vertical bars indicate mean \pm one standard deviation across individuals or randomizations. Paths are grouped according to the 1st up to the 5th shortest path between region pairs, showing that relay communication is not limited to the 1st shortest path only. **(b)** Group-average parallel communication score (PCS) matrices representing PCSs between every pair of brain regions, averaged across individuals. For each species, brain regions are organized according to meaningful functional circuits which are highlighted by black squares along the matrices' diagonals and by color-coded bars (Online Methods). On the right, the histograms of the average PCS scores across region pairs highlight an evolutionary gap between humans, with higher PCSs and presence of parallel communication, and macaques and mice, with lower PCSs and mainly selective information processing. Median [5-, 95-percentile] PCS values for each species are reported atop each histogram. VIS=visual; SM=somatomotor; DA=dorsal attention; VA=ventral attention; L=limbic; CN=control; DMN=default mode; BR=barrel; AD=auditory networks.

Supplementary Figure 2 Parallel communication gap between humans and other mammalian species. Results from pairwise statistical comparisons between group-average parallel communication scores (PCSs) of humans, macaques and mice (all main and replication datasets included). Main (non-anesthetized) and replication (anesthetized) datasets are reported on rows and columns; an asterisk next to the dataset name indicates that animals were anesthetized. The colorbar represents group-difference effect size, computed as the difference of the medians normalized by the pooled median absolute deviation of the two groups. Violet tones indicate that the row dataset has larger PCSs than the column dataset; orange tones indicate that the row dataset has smaller PCSs than the column dataset. White asterisks in the matrix indicate p -values $< .05/15$ (Mann-Whitney U tests, Bonferroni corrected).

Besides group-average distributions, we investigated individual-level parallel communication scores. To this end, we assessed for each experimental subject the average PCS within unimodal brain systems, within transmodal brain systems, and between unimodal and transmodal brain systems ('cross-modal' connections) (Fig. 3c). We compared these individual-level scores between every pair of datasets, including both anesthetized and non-anesthetized animals. The cross-species gap between human vs macaques and mice remained strongly significant at the individual subject level, with effects localized in cross-modal connections independently from animals' anesthesia (SI Fig. 5). We found no major difference between anesthetized and non-anesthetized macaques or mice, although the amount of relay communication was slightly larger in non-anesthetized mice (m-GG) compared to one of the two anesthetized murine datasets (m-AD3) in both transmodal and cross-modal systems (SI Fig. 5, central and bottom rows).

Figure 3. Relay communication strategies reflect the functional organization of mammalian brains. (a) Cortical distributions of relay communication, quantified as the average PCS of each brain region with the rest of the brain network (first row: human, *fsaverage6* cortical surface; second row: macaque, *F99* template; third row: mouse, *ABI* template). For each species, the light yellow-to-brown colormap is scaled between the 5th and 95th percentiles of the cortical values. On the right, the average nodal communication scores per brain system are represented in the bar plots. (b) Average PCSs within and between brain systems, for humans, macaques and mice. Brain systems have been organized into unimodal/multimodal regions (upper-left black square) and transmodal regions (lower-right black square). (c) Average PCSs between unimodal systems, between transmodal systems, and between unimodal and transmodal systems (cross-modal communication) for individual experimental subjects. In the box plots, each dot represents a subject; vertical bars indicate mean \pm standard deviation; notch bars indicate median and 1st-3rd quartiles; shaded areas indicate 1st-99th

percentiles. Kruskal Wallis *p*-values for within-species comparisons are reported, testing the null hypothesis of whether unimodal, transmodal and cross-modal PCS scores originate from the same distribution. All datasets include only non-anesthetized experimental subjects. Control = executive-control; Dorsal attn. = dorsal attention; ventral attn. = ventral attention; uni = unimodal/multimodal; trans = transmodal.

Supplementary Figure 5. Individual parallel communication scores across species and brain systems. Average PCSs between unimodal systems (top row), between transmodal systems (middle row), and interconnecting unimodal and transmodal systems ('cross-modal' region pairs; bottom row) for individual experimental subjects of the three mammalian species (humans: blue; macaques: green; mice: brick red), and for all main and replication datasets. Statistically significant pairwise comparisons are indicated with horizontal black lines (Mann-

Whitney U tests; $p < .05 / 15$, Bonferroni-corrected multiple comparisons). Datasets including anesthetized animals are indicated with a gray-shaded rectangle and an asterisk. In the box plots, each dot represents an individual; vertical bars indicate mean \pm standard deviation; notch bars indicate median and 1st-3rd quartiles; shaded areas indicate 1st-99th percentiles.

Finally, please note the analyses reported in SI Fig. 16 and 17, where we test possible dataset-specific biases (including the effect of anesthesia) at the functional network level. Results showed a generally preserved topology of the communication patterns within species, regardless of anesthesia (compare SI Fig. 16 and 17 with the updated Fig. 3 reported above).

Taken together, our group-level and individual-level results indicate the presence of a cross-species gap between humans and other mammalian animals, with human brain communication tailored towards parallel information transmission and other mammalian brains tailored towards selective information transmission. Phylogenetically older species (macaques and mice) demonstrate more developed relay communication within unimodal systems for lower-order processing between somatosensory and attention regions. Conversely, the human brain is characterized by stronger parallel communication that serves as the main neural processing stream between unimodal and transmodal areas.

Our results do not allow us to draw major conclusions on the effect of anesthesia on brain communication processes. In general, non-anesthetized animals seem to have a larger amount of relay communication than anesthetized ones. However, these minor effects may be affected by other factors including differences in acquisition protocols and type of anesthesia. For example, different anesthetics have been shown to have different modulatory effects on fMRI brain activity (Grandjean et al., 2014). Further analyses are therefore needed to elucidate the effect of anesthesia on brain communication patterns. Particularly, it will be interesting to investigate functional recordings from the same experimental subjects, assessed before and under the effect of anesthesia. These limitations are now discussed in the main text, which now reads:

“When considering fMRI data from awake and anesthetized animals, we found similar cortical distributions of parallel communication patterns, with unimodal regions dominated by selective information processing. This finding is in line with the observation that resting state networks are globally preserved in conscious and unconscious states^{18,48}. Yet, our results do not allow us to draw major conclusions on the effect of anesthesia on brain communication processes. In general, non-anesthetized animals seem to have a larger amount of relay communication than anesthetized ones. A shift of the communication regime toward more abundant and (partially) parallelized polysynaptic information transmission may mechanistically support functional integration, inter-network cross-talk, and rich functional repertoires departing from the underlying monosynaptic connectivity constraints, which have been repeatedly observed in awake primates and mice compared to the anesthetized ones^{18,48,49}. Nevertheless, these minor effects may be affected by other factors including differences in acquisition protocols and type of anesthesia. For example, different anesthetics have been shown to have different modulatory

effects on fMRI brain activity⁵⁰. Further analyses are therefore needed to elucidate the effect of anesthesia on brain communication processes. Particularly, it will be interesting to investigate functional recordings from the same experimental subjects, assessed before and under the effect of anesthesia.”

Following up on the Reviewer’s remark, we do acknowledge that the concept of “evolutionary gradient” could be too strong for the reported analyses. However, we also want to point out that the gap between parallel communication scores in humans vs animals is strong, significant and consistent across all the datasets evaluated, both at the group level and at the individual subjects’ level. Therefore, we have reworded the term “evolutionary gradient” with the more appropriate “cross-species gap” between humans and other mammalian species in several parts of the manuscript, including the title of the manuscript (which now reads “Information transmission in mammalian brain networks”) and Fig. 1. The updated Fig. 1 is reported below:

Figure 1 Identifying communication pathways in macroscale brain networks. ... (e) Parallel communication scores are investigated across mammalian species, highlighting a spectrum of communication strategies from selective information processing (light yellow; low PCS), to parallel information processing (dark brown; high PCS). Particularly, our work highlights a parallel relay communication gap between humans and animals (macaques, mice), with human communication mechanisms tailored towards parallel communication.

Finally, while I acknowledge the difficulties (and ambition) of trying to analyze datasets from

many different sources, the differences in MRI scanning protocols (e.g., TR length) give many degrees of freedom that were not fully addressed. I detail these concerns (and others) below.

Please find below our detailed responses to your concerns.

Major

* A major concern is that the results do not clearly reflect the hypothesized differences of PCS across mammalian species when controlling for anesthesia. Specifically, unanesthetized mice had greater PCS than unanesthetized monkeys (Supp Fig. 14d vs f). This appears to be inconsistent with the claims made in the Results section that reference the supplementary analysis (line 400). This appears to be a misleading mischaracterization of supplementary data. In general, I think the unanesthetized results should all be presented in the main text, and the anesthetized results placed in the supplementary (to avoid any confusion).

See our detailed answer above on the effect of anesthesia on the PCS scores. We have followed the Reviewer's suggestion and now placed the unanesthetized results in the main text, and the anesthetized results in the Supplementary, updating all the Figures.

* I'm not an expert in evolutionary biology, but is it common to phrase the transition from mouse to monkeys to humans as "evolutionary complexity"? From my naive knowledge on evolution, all three species are highly specialized for their current (contemporary) environments, and I'm curious to know if it's common to think of humans as a "more evolved form" of a mouse, for example. This seems to be an oversimplification. Otherwise, citations are warranted.

We agree with the Reviewer. All these species are complex animals and the wording "evolutionary complexity" might be misleading. Note that we have smoothed out the claims of evolutionary gradient (see reply above) and, along those lines, we have removed the term "evolutionary complexity" from the text.

* The choice of 5 shortest structural paths seems somewhat arbitrary to me. If I understand correctly, Supplementary Fig. 1b seems to indicate that a majority of the shortest paths are only 1-step connections. It follows that most likely, the 5 shortest connections will likely be 1-step connections. What is the significance of DPI of a 1-step connection? Can it truly be considered 'parallel' if this analysis is reduced to if the top 5 1-step connections obey the DPI (whereas the schematic in Fig. 1 appears to be an illustration for 2-step connections)?

We thank the Reviewer for bringing up this aspect. Indeed, the DPI framework concerns multi-step paths only, since the data processing inequality cannot be assessed on 1-step connections. Therefore, our results concern multi-step paths only, which is in line with the concept of parallel communication. Note that the considered structural networks, which were built from diffusion MRI and track tracing data, do not include multiple 'parallel' 1-step

connections. Each pair of brain regions can be connected by at most one 1-step connection. This is now better explained in the text, Online Methods section:

“..That is, we aim to characterize the presence of memoryless information transmission processes, with information decay along the path measured through mutual information values. Note that the DPI framework concerns multi-step paths only, since the data processing inequality cannot be assessed on 1-step connections. Therefore, our results concern multi-step paths only, which is in line with the concept of parallel communication. Note also that the considered structural networks, which were built from diffusion MRI and track tracing data, do not include multiple ‘parallel’ 1-step connections. Each pair of brain regions can be connected by at most one 1-step connection.”

Consistently, SI Fig. 1b shows that only a minority of the investigated k -shortest paths are 1-step connections (blue bars). The proportions of 1-step connections over the whole k -shortest path ensembles investigated in the three species are now specified in SI Fig. 1 legend for clarity:

Supplementary Figure 1. Cross-species brain structural and functional data. ... (b) Percentage of k -shortest paths of length 1 (blue), 2 (orange), 3 (yellow), 4 (violet), or 5 (green) steps for the different matrices. Note the strong similarity between species. 1-step paths

represent a minority of the overall k -shortest path ensembles in the three species (9%, 11% and 10%, respectively).

The choice of the parameter k for the k -shortest path assessment was done based on previous work on path ensembles in brain networks, where we showed that considering only the first shortest paths excludes a significant fraction of network connections (up to 80%) from participating in any structural multi-step path (Avena-Koenigsberger et al., 2017). This result reinforces the idea that considering only the first shortest paths in brain communication models is strongly reductive, since all brain network connections are expected to participate in communication paths. When investigating the fraction of network connections participating in at least one multi-step path as a function of k , we showed that this fraction increases with k and rapidly converges to a plateau at $k \sim 5$ (Avena-Koenigsberger et al., 2017), which motivated our choice of the parameter k . In our dataset, when considering the first 5-shortest paths between any pair of brain regions, 91% / 87% / 94% of network connections are used in at least one multi-step path for humans / macaques / mice. The rationale behind the choice of the parameters k is now better explained in the Online Methods section:

“ K -shortest path ensembles identify meaningful trade-offs between efficiency and resiliency for putative communication processes in brain networks²⁴. The choice of k was dictated by the fact that, for $k=5$, all edges of the structural brain network participate in at least one k -shortest path²⁴. In fact, previous results showed that, when investigating the fraction of network connections participating in at least one multi-step path as a function of k , this fraction increases with k and rapidly converges to a plateau at $k \sim 5$ ²⁴. This finding was confirmed in our study where, when considering the first 5-shortest paths between any pair of brain regions, we found that 91% / 87% / 94% of network connections were used in at least one multi-step path for humans / macaques / mice respectively. ”

* Another question I had about the SC matrix was that it appears that SC matrices were binarized, and then the weights were modified by their Euclidean distances. However, wouldn't this bias the shortest paths to be very close range connections? Would any long range connections be included? Perhaps it could be helpful to provide some examples of what long range connections are included for a few example regions.

We checked whether all brain connections participate in the k -shortest paths, independently from their physical length. We found that, for all species, the large majority of brain connections are included in at least one k -shortest path (see also our answer to your previous comment), and therefore long range connections are included in our analyses.

That being said, the Reviewer is right in that, in our model, short range connections are preferred to long ones - this is the principle behind shortest path routing in complex networks. From a neurobiological perspective, neural signal transmission through physically shorter axons is less costly in terms of metabolic resources and may therefore be preferred to longer

connections (Bullmore and Sporns, 2012). This is now mentioned in the Online Methods for clarity:

“Group-representative structural connectivity matrices with comparable number of brain regions were derived from diffusion MRI and tract tracing data, and weighted by the Euclidean distance between connected regions since, from a neurobiological perspective, neural signal transmission through physically shorter axons is less costly in terms of metabolic resources and may be preferred to longer connections⁶⁾”

When considering all the 5-shortest paths investigated for each species, we observe an expected inverse relationship between the number of k -shortest paths a white matter connection participates in, and the connection length (quantified as the Euclidean distance between region pairs' centroids). However, connections participating in a non-negligible number of k -shortest paths present a broad range of lengths, as highlighted by the triangular patterns shown in figure:

Figure description: Number of k -shortest paths a white matter connection participates in (y-axis), plotted as a function of the connection length (x-axis), for the three species (human, macaque, mouse). The numbers of k -shortest paths (y-axis) are represented in logarithm (base 10) scale.

Taken together, these data indicate that long range connections are well included in our brain communication analyses.

* Evaluating and comparing relay communication patterns that are unique to individuals. How were the fMRI recordings split? Were independent imaging sessions tested apart? Or were they data points within the same session? It wasn't immediately clear from the Methods, but for statistical independence sessions/data should be assessed from independent scans. Otherwise individual fingerprinting would be confounded if data were taken from the same scan session.

Thanks for raising this important concern. Indeed, a common problem in cross-species studies is that it is very difficult to acquire test/retest sessions in animal cohorts, for several reasons. One potential workaround to this issue might be to cut the resting state time series in half (as originally proposed in Amico and Goni, Scientific Reports 2018, see SI Fig. 3 of that article). This has the benefit of removing the scanner and acquisition noise, which is usually a major confound in connectome identification (Bari et al., NeuroImage 2019). However, this

workaround comes at the cost of looking at “within session” fingerprinting, which focuses more on the temporal stability aspect of the communication patterns rather than the “standard” between-session identification. Any researcher who wants to replicate the same analysis should be aware of this detail. Nonetheless, we believe it is noteworthy that humans could be better identified than animals solely on the basis of their (within-scanner) parallel communication profiles. We have now clarified this aspect in the Discussion, that now reads:

“Subject specificity of communication patterns was assessed on relatively short time series (250 to 300 time points). Recent studies have shown that one does not need long fMRI scans to achieve high test-retest reliability (see Figure 5 in ³⁹, or Figure 2 in ⁴⁸). Based on this recent evidence, it appears that ~100 fMRI volumes would be enough to achieve good success rates and identifiability scores in humans. Furthermore, one common problem in cross-species studies is that it is usually very difficult to acquire test/retest sessions in the different cohorts. One potential workaround to this issue might be to cut the resting state time series in half (as originally proposed in ³⁹, see Supplementary Figure 3). This has the benefit of removing the scanner and acquisition noise, which is usually a major confound in connectome identification⁴⁹. However, it comes at the cost of looking at within-session fingerprinting, hence focusing more on the temporal stability aspect of the communication pattern, rather than on standard between-session identification. Nonetheless, it is noteworthy that humans could be better identified than animals solely on the basis of their (within-scanner) parallel communication profiles. Future studies should explore how the results change when considering multiple and longer sessions, whenever available, to estimate cross-species communication fingerprints.”

* I would encourage the authors to be a bit more careful with terminology. Throughout the manuscript, they often equate PCS with the brain’s information communication strategy (e.g., line 420 (Discussion)). However, the measure of PCS is a information communication strategy that is imposed by the author (experimenter-derived measure), rather than a strategy that is actually implemented by the brains. The authors should clarify this in the Introduction/Discussion.

We agree with the Reviewer. We have corrected the equation “DPI/PCS = brain communication strategy” throughout the manuscript, highlighting the fact that our framework represents one (possible) model to characterize fMRI signals in relation to brain structure - please, also see our answer to the Reviewer’s first comment. We have changed the wording ‘communication strategy’ with ‘communication model / modeling’ throughout the manuscript. For example, the first paragraph of the Discussion (line ~420) now reads:

“In vivo measurements of brain structure and activity are providing us with windows of opportunities for modeling communication in brain networks, across different animal species. We propose here to bring a piece to this puzzle, by investigating the link between communication models in large-scale brain networks, on the one side, and the phylogenetic level of mammals’ brain functions, on the other.”

* The measurement of SNR seems to be confounded by the differences in TR. Since (according to the methods) SNR is just the ratio of low v. high freq power contents, determined by the half of the Nyquist frequency, then SNR measurements across scans with different TR lengths are likely incomparable.

The Reviewer is correct. We considered a rudimental measure of SNR, defined as the ratio between low vs high frequency power contents from a median split of the spectrum. The SNR computed in this way is indeed affected by the TR, but please note that we never compared SNRs between datasets with different TRs. We only used it to look at within-species associations between SNR and PCSs (which were non-significant, see our Supplementary Fig. 11). In this sense, different TRs do not affect our conclusions - i.e., parallel communication scores are not driven by regional SNR.

Minor

* The mismatch between using group-level SC matrices versus individual-level FC (or mutual information) analyses seems problematic. I appreciate the difficulties in estimating individual SCs due to high variance of diffusion scans, but this seems like it should be addressed as a limitation. Moreover, this might suggest that the individual fingerprinting of PCS (Fig. 4) derives entirely from the mutual information analysis, rather than the combination of mutual information + SC + DPI analysis. Does fingerprinting of PCS provide something over and above just functional fingerprinting (using, for example, mutual information)?

The Reviewer is right. We used group-average rather than individual-level SC matrices. Note that small animal (e.g. mouse) diffusion tractography is particularly challenging and tract-tracing data remain a reference in the field (see Arefin et al., 2023 as an example).

The analyses reported in Fig. 4 are meant to corroborate the neuroscientific relevance of the new measure introduced in this work—the parallel communication score — by showing its specificity to individual subjects. This is particularly interesting considering that the PCS, which can take integer values between 0 and 5, is a highly compressed measure compared to mutual information, which can take real values between 0 and 1. Our analyses do not allow us to draw any conclusions on which brain dimension (structural connectivity, functional connectivity, parallel communication among others) is the most subject-specific or most appropriate for a fingerprinting analysis *per se*, which is beyond the scope of this work.

These aspects are now addressed as limitations in the Discussion:

“Finally, the analyses reported in Fig. 4 showed that the parallel communication score showed specificity to individuals: This is particularly interesting considering that the PCS, which can take integer values between 0 and 5, is a highly compressed measure with respect to functional connectomes. However, our analyses do not allow us to draw any conclusions on which brain

dimension (structural connectivity, functional connectivity, parallel communication among others) is the most subject-specific or most appropriate for a fingerprinting analysis per se, which is beyond the scope of this work.”

* How would PCS be assessed if using just linear correlation rather than mutual information? What is the rationale for using mutual information, given that correlations are just the linear equivalent of MI?

The Reviewer is right in that, when probing fMRI data, linear correlation and mutual information values are strongly related, as it has been previously assessed (Hlinka et al., 2011). However, in this work we used mutual information since, in information theoretical terms, it directly quantifies the amount of information shared between two random variables and it is intrinsic to the Data Processing Inequality statement. Our choice is therefore instrumental to our information theory framework.

* I wonder if species-specific PCS differences across unimodal/transmodal areas could be in part explained by differences in the structure-function tethering across those species? The specific work that comes to mind is: (Baum et al., 2020; Vázquez-Rodríguez et al., 2019).

We thank the Reviewer for this interesting perspective, which we now discuss in the manuscript:

“Which evolutionary mechanisms may have promoted a higher involvement of parallel communication in humans? According to the tethering hypothesis proposed by Buckner and Krienen²¹, the fast cortical expansion of transmodal regions in humans led to the untethering of these regions from developmental anchor points. In parallel, humans exhibit a protracted development of white matter connections over childhood and a progressive structure-function untethering in transmodal regions compared to other primates²⁰. Cortical expansion and developmental trajectories in humans may have therefore allowed transmodal regions to develop unique cytoarchitectonic⁴³ and connectional³¹ fingerprints, unbounded from the more rigid hierarchical architecture of unimodal systems³⁸. The same processes may have also favored the development of new information transmission pathways (parallel communication) to bridge hierarchical unimodal and distributed transmodal regions. Indeed, in humans we observed the largest parallel communication scores in regions that underwent the largest cortical expansion across evolution and the largest changes of structure-function coupling across development, including fronto-parietal association cortices and precune⁴⁴. Consistently, it has been shown that the level of structure-function coupling in humans is highly heterogeneous across the cortical mantle and reaches a minimum in transmodal regions at adulthood, which may be critical for the maturation of complex cognitive functions^{20,45}.”

* Is merely shuffling the mutual information matrix appropriate as a permutation/null test (e.g., in Fig. 2)? Since both fMRI and the SC matrix are both confounded by spatial autocorrelation, my

sense is that arbitrarily shuffling the mutual information matrix without preserving any spatial autocorrelation may not be the appropriate null to test whether there are true differences in PCS across unimodal-transmodal areas.

We appreciated the comment of the Reviewer, which led us to perform additional analyses. Inspired by the recent work of Burt and colleagues (Burt et al., 2020), we develop an additional null model for mutual information (MI)-based functional connectomes, which preserves their spatial autocorrelation with respect to the underlying distance-weighted structural connectivity matrix. Specifically, similarly to the brainSMASH algorithm (Burt et al., 2020), we first quantified the level of spatial autocorrelation in the original individual MI matrices with a variogram. This is the target variogram for the output null MI matrices. Then, the original individual MI matrix is randomly permuted. Spatial autocorrelation among the samples is reintroduced by smoothing the permuted map with a distance-dependent kernel (similarly to BrainSMASH, here we employed an exponentially decaying gaussian kernel). Afterwards, the smoothed MI matrix's variogram is computed and then regressed onto the variogram for the original MI matrix. The goodness-of-fit is quantified by computing the sum of squared error (SSE) in the null vs. original variogram fit. Finally, for every individual, we picked the null MI matrix with the best goodness-of-fit over 100 permutation runs.

When comparing experimental data against this spatial-autocorrelation preserving null model, we found that the latter could not explain alone the differences in PCS across unimodal-transmodal areas. This confirms once again that our findings are not trivially driven by spatial and/or topological properties of the brain networks under investigation, but rather by the species-specific functional-structural interplay investigated through DPI.

These analyses are reported in a new Supplementary Fig. 20, which we also report below for completeness.

Supplementary Figure 20. Comparison between real data and spatially structured physiological noise null model. Left: drawings of human, macaque, and mouse brains; each row in the figure corresponds to one species. **(a)** Box plots representing the percentage of short paths in individual brain networks used for relayed communication (i.e., respecting the DPI). Each colored dot represents an individual. Gray dots represent null distributions obtained by shuffling individual mutual information values while preserving network-level autocorrelation patterns (i.e., while preserving the mutual information variogram); the brain network structural topology was unchanged (Online Methods). Circles and vertical bars indicate mean \pm one standard deviation across individuals and randomizations. Paths are grouped according to the 1st up to the 5th shortest path between region pairs. **(b)** Histograms of group-average PCSs obtained from real data (blue) and spatially structured noise (pink). For all the species, the real PCS histograms significantly differ from the null ones, indicating that parallel communication levels are not the trivial byproduct of the network-level autocorrelation present in functional brain data. **(c)** Average PCSs between unimodal systems, between transmodal systems, and between unimodal and transmodal systems (cross-modal communication) for individual experimental subjects. In the box plots, each dot represents a subject individual; vertical bars indicate mean \pm standard deviation; notch bars indicate median and 1st-3rd quartiles; shaded areas indicate 1st-99th percentiles. Values from the spatially structured null model are reported in the same panels as gray box plots. For all species and brain systems, the real individual PCS significantly differ from the null ones, indicating that the network-level spatial autocorrelation of functional signals cannot explain system-dependent spatial communication patterns.

* In supplementary Figure 17, what do the p-values in the top-left corner of each graph reflect? Same for Figure 3 and related figures. The specific statistical test is reported, but not what is being compared.

We apologize for the lack of clarity. The figure legends now read:

... Kruskal Wallis p-values for within-species comparisons are reported, testing the null hypothesis that unimodal, transmodal and cross-modal PCS scores originate from the same distribution.

References

Baum, G.L., Cui, Z., Roalf, D.R., Ciric, R., Betzel, R.F., Larsen, B., Cieslak, M., Cook, P.A., Xia, C.H., Moore, T.M., Ruparel, K., Oathes, D.J., Alexander-Bloch, A.F., Shinohara, R.T., Raznahan, A., Gur, R.E., Gur, R.C., Bassett, D.S., Satterthwaite, T.D., 2020. Development of structure–function coupling in human brain networks during youth. *Proc. Natl. Acad. Sci.* 117, 771–778. <https://doi.org/10.1073/pnas.1912034117>

Vázquez-Rodríguez, B., Suárez, L.E., Markello, R.D., Shafiei, G., Paquola, C., Hagmann, P., Heuvel, M.P. van den, Bernhardt, B.C., Spreng, R.N., Misic, B., 2019. Gradients of structure–function tethering across neocortex. *Proc. Natl. Acad. Sci.* 116, 21219–21227. <https://doi.org/10.1073/pnas.1903403116>

References

Amico, E., Goñi, J., 2018. The quest for identifiability in human functional connectomes. *Scientific Reports* 8:8254

Arefin, T.M., Lee, C.H., Liang, Z., et al., 2023. Towards reliable reconstruction of the mouse brain corticothalamic connectivity using diffusion MRI. *NeuroImage* 273:120111

Avena-Koenigsberger, A., Misic, B., Hawkins, R.X.D., et al., 2017. Path ensembles and a tradeoff between communication efficiency and resilience in the human connectome. *Brain Structure and Function* 222:603-618

Avena-Koenigsberger, A., Misic, B., Sporns, O., 2018. Communication dynamics in complex brain networks. *Nat Rev Neurosci* 19: 17–33

Bari, S., Amico, E., Vike, N., et al., 2019. Uncovering multi-site identifiability based on resting-state functional connectomes. *NeuroImage* 202:115967

Bullmore E., Sporns O., 2012. The economy of brain network organization. *Nature Review Neuroscience* 13:336-349

Burt, J.B., Helmer, M., Shinn, M., et al., 2020. Generative modeling of brain maps with spatial autocorrelation. *NeuroImage* 220: 117038

Grandjean, J., Schroeter, A., Batata, I., Rudin, M., 2014. Optimization of anesthesia protocol for resting-state fMRI in mice based on differential effects of anesthetics on functional connectivity patterns. *NeuroImage* 102(2): 838-847.

Hlinka, J., Paluš, M., Vejmelka, M., et al., 2011. Functional connectivity in resting-state fMRI: Is linear correlation sufficient? *NeuroImage* 54(3): 2218-2225

Luppi, A.I., Craig, M.M., Pappas, I., et al., 2022. A synergistic core for human brain evolution and cognition. *Nature Neuroscience* 25:771-782

Santoro, A., Battiston, F., Petri, G., Amico E., 2023. Higher-order organization of multivariate time series. *Nature Physics* 19:221-229

Reviewer #2 (Remarks to the Author):

The manuscript “The evolution of information transmission in mammalian brain networks“ by Griffa et al describes a novel methodology to quantify functional connectivity between brain regions based on the number of parallel structural connections that are used for information transmission. They use an information theoretic approach assuming that mutual information between a signal observed in a source and along the pathways should be decreasing monotonically, and that the pathway itself is memoryless.

First, the structural connectome among brain regions is derived based on diffusion tensor imaging data. Then, connections between brain regions are evaluated based on paths with a defined length through the structural network.

Overall this approach adds a new perspective to the assessment of functional networks. Even though the authors suggest several insights based on the comparison of mouse, macaque, and human brain, the primary contribution is methodological.

In the following a few points:

It would facilitate reading, if the paper would start with describing the claim of counting parallel structural connections that are used for signal transmission as observed in functional magnetic resonance imaging (fMRI). After that, introduce the information theoretic approach used to evaluate this. Currently for a long initial stretch, the only point readers learn about the novelty is that it is an “information theoretic approach“.

We thank the Reviewer for this comment. Following the Reviewer’s suggestion, we have restructured the Introduction to better (and earlier) highlight the rationale of counting parallel communication pathways. We report below the second paragraph of the updated Introduction (please, see our answer to the next comment).

Provide a motivation what exact insight this adds to our understanding of brain architecture and its evolution. What are many parallel connections associated with? Are there prior findings, or hypotheses regarding the functional brain architecture that explain the rationale? What are the “different communication strategies“ associated with parallel paths used at the same time. Please explain, and clarify this - as one of the main claims is regarding evolution, this is an important part. Maybe one can expand figure 1 to explain this overall concept.

Recent evidence indicates that communication cannot be explained with simple shortest path routing (Seguin et al., 2020; Goñi et al. 2014, Avena-Koenigsberger et al., 2017). Most recent work also showed that more advanced communication models provide better association with functional brain dynamics (Seguin et al., 2023), highlighting the relevance of multiple communication pathways (beyond the shortest path) to brain communication models. To the best of our knowledge, ours is the first work that investigates information transmission between

region pairs through multiple communication pathways by merging structural and functional information in mammalian brain networks.

The idea of looking at parallel communication came from different considerations. First, selective information routing through the shortest paths only is not a satisfactory model of brain communication in that it does not well explain the observed functional MRI recordings (Goñi et al., 2014). Second, information routing through multiple, parallel communication pathways has been suggested to increase transmission fidelity, robustness and resilience to brain damage while achieving a reasonable trade-off between communication efficiency and metabolic expenditure (Avena-Koenigsberger et al., 2018; Zhou et al., 2023). Third, multiple communication channels may be used together or separately at different moments in time to support changing internal and external representations and complex functions including higher-order cognition (Luppi et al., 2022). In this sense, a compelling hypothesis is that parallel communication developed across mammalian evolution to support cognitive tasks of increasing complexity.

Different measures of parallel communication have been proposed for the structural connectome (such as path ensembles and communicability (Avena-Koenigsberger et al., 2017; Estrada et al., 2008)) but do not embed brain functional information. Having a well-grounded measurement of information transmission through selective and parallel pathways can help in disentangling differences in communication mechanisms across mammalian species.

We elaborate more on these aspects and motivations in the Introduction of our manuscript, which now reads:

“Brain networks of several mammalian and simpler species have short structural path length^{21,22} at the price of a relatively high wiring cost⁵, suggesting that polysynaptic shortest paths contribute to efficient communication in brain networks and have been selected throughout evolution despite their high wiring cost. Yet, models of selective communication through shortest paths only explain a limited portion of functional connectivity¹⁰ and exclude a large fraction of brain network connections and near-optimal alternative pathways from the communication process²³, pointing out the relevance of multiple pathways to brain communication models^{1,16,17}. Indeed, in many real-world systems, information transmission unfolds through numerous alternative pathways according to a parallel communication scheme²⁹. In the brain, parallel communication may increase transmission fidelity, robustness and resilience to brain damage¹ while achieving a reasonable trade-off between communication efficiency and metabolic expenditure¹⁸. Moreover, multiple communication channels may be used together or separately at different moments in time to support changing internal and external representations and complex functions including higher-order cognition¹⁹. In this sense, a compelling hypothesis is that parallel communication may have evolved across mammalian evolution to support cognitive tasks of increasing complexity. Nonetheless, the information transmission mechanisms implemented in mammalian brain networks and, particularly, the relative contribution of single-pathway (‘selective’) versus multiple-pathway (‘parallel’) communication are, to date, largely unknown.”

Is it correct that the paths run along the cortex? That is, they are several hops from cortical region to cortical region? I assume this is the case, since your results are based on fMRI, but it is worth clarifying early.

The Reviewer is right in that the considered paths are made of several hops from cortical region to cortical region. However, two regions do not have to be neighbors in space to be part of a path. They only have to be neighbors in the brain network built from tractography and tract tracing data. This is now better specified in the Results section:

“Given a structural network representing the white matter wiring of the brain, we first identify sets of short polysynaptic paths connecting each pair of brain regions (Fig. 1a). These structural paths are made of several hops from cortical region to neighborhood cortical region in the structural connectome.”

Figure 2 is overall helpful, but it would be better, if you provide a visualization of the brain regions corresponding to the block-diagonal segments.

We thank the Reviewer for this suggestion. We have updated Figure 2 including color-coded bars and brain systems' abbreviations to facilitate the interpretation of the block-diagonal segments in the parallel communication score matrices. The colors of the color-coded bars are consistent with the visualization of brain regions provided in Supplementary Fig. 3 and 4. We report below the updated Fig. 2 for completeness.

Figure 2. Parallel communication gap across mammalian species. Left: drawing of human, macaque, and mouse brains; each row in the figure corresponds to one species. **(a)** Box plots representing the percentage of short paths in individual brain networks used for relayed communication (i.e., respecting the DPI). Each colored dot represents an individual; gray dots represent species-specific null distributions obtained from permutation of mutual information values (Online Methods); circles and vertical bars indicate mean \pm one standard deviation across individuals or randomizations. Paths are grouped according to the 1st up to the 5th shortest path between region pairs, showing that relay communication is not limited to the 1st shortest path only. **(b)** Group-average parallel communication score (PCS) matrices representing PCSs between every pair of brain regions, averaged across individuals. For each species, brain regions are organized according to meaningful functional circuits which are highlighted by black squares along the matrices' diagonals and by color-coded bars (Online Methods). On the right, the histograms of the average PCS scores across region pairs highlight an evolutionary gap between humans, with higher PCSs and presence of parallel communication, and macaques and mice, with lower PCSs and mainly selective information processing. Median [5-, 95-percentile] PCS values for each species are reported atop each histogram. VIS=visual; SM=somatomotor; DA=dorsal attention; VA=ventral attention; L=limbic; CN=control; DMN=default mode; BR=barrel; AD=auditory networks.

Please clarify the term „relay communication density“. Is there some context that can help understand its significance in this paper?

We apologize for the lack of clarity. ‘Relay communication density’ is a new measure we

introduce in this paper. It represents the percentage of k -shortest paths (i.e., of structural paths derived from the structural brain network) that respect the data processing inequality (DPI). Conceptually, this measure indicates the amount of relay communication taking place in the different brains (individuals and species). We elaborate on this in the updated Results section:

“The parallel communication density is a new measure we introduce in this work which indicates the amount of parallel communication that takes place in individual brains.”

If I understand correctly, then the conclusion of Figure 2 is one of the main conclusions of the paper: communication becomes more parallel when comparing mouse to macaque to human. Is it possible that this is in part an artefact of increased brain size, corresponding increased relative DTI resolution, and therefore overall higher numbers of structural connections?

We understand the concern of the Reviewer and we now highlight this aspect in the Discussion. However, we would like to point out that the number of considered structural paths was the very same for the three species (5 paths per region pair). Moreover, the number of direct structural connections was approximately the same for the three species (network densities equal to 0.47 / 0.56 / 0.48 in humans, macaques, mice; Supplementary Fig. 1).

Please also note that our results in Supplementary Fig. 1, 8 and 9 should already exclude this possibility, since they show that (i) the proportion of 2-step, 3-step, and 4-step short structural paths is similar across species, indicating a comparable availability of structural k -shortest paths; and (ii) the gap of increasing parallel communication from mice to humans is stable when normalizing data with respect to null models specific to each species, suggesting independence of our results from trivial white matter connectivity and SNR differences.

Nonetheless, we have performed the following additional verification, to check whether white matter complexity is associated with PCSs. Our group-averaged structural connectivity (SC) matrix in humans was based on connections present in 100% of the cohort. We have tuned this threshold, in order to let more connections in, hence increasing the “white matter complexity” of human SC, going from our original threshold (100%), as low as 50% (i.e., we allow a connection when present in half of the cohort), in decreasing steps of 10%. We then recomputed PCSs at each step, to test its dependence from white matter complexity. Our results, shown in the new Supplementary Fig. 12 below, demonstrate that the PCS does not depend on the threshold selection for the group representative SC, confirming once again that our results can not be trivially explained by white matter complexity and brain size alone.

Supplementary Figure 12. Dependence of Parallel Communication Scores (PCS) on white matter complexity. The box plot depicts individual PCS (y-axis) when changing human group representative structural connectome, by changing the consensus threshold from 50% (allowing a connection when present in half the cohort) to 100% (the value chosen for main text results), in 10% increasing steps (x-axis). Note how Individual PCS are stable across the choice of the threshold, confirming that thresholding of the structural connectome cannot explain the brain communication patterns found.

The authors perform analysis to create a null-connectivity model by randomly permuting the assignment of time courses to cortical regions - that makes sense. The number of parallel connections remains significant after this (line 371). However, this statement does not extend to the differences in parallelism between the three species. The results in Figure 7 suggest that after this z-score normalization of parallel transmission pathways, the 3 species are almost the same. Please clarify: is there still a significant gradient of increasing parallelism, when controlling for the number of connections? This is a central claim of the paper, and needs proper testing.

The presence of a parallel communication gap between humans and phylogenetically older mammalian species is the central claim of this paper, and we have performed additional testing in support of this claim.

First, following the suggestion of Reviewer 1, we now directly compare human data with non-anesthetized macaques' and mice' data. This is to avoid any possible confounding factor related to anesthesia. We have moved all results from anesthetized animals to the Supplementary

Information.

We report in the updated Fig. 2 and updated Supplementary Fig. 8 the distributions of the group-average parallel communication scores (PCSs) from non-anesthetized experimental subjects. The distributions were pairwise statistically different both when considering the raw PCSs (two-sample Kolmogorov-Smirnov tests; human-macaque: $D_{4950,3321} = 0.180$, $p < 10^{-55}$; human-mouse: $D_{4950,2145} = 0.144$, $p < 10^{-27}$; macaque-mouse: $D_{3321,2145} = 0.076$, $p < 10^{-6}$) and when considering z-scored PCSs ($D_{2087,794} = 0.309$, $p < 10^{-47}$; human-mouse: $D_{2087,788} = 0.297$, $p < 10^{-43}$; macaque-mouse: $D_{794,788} = 0.099$, $p < 10^{-3}$), although the distances between macaque and mouse distributions were minor. The median PCS was larger in humans compared to both macaques and mice (median [5-, 95-percentile] across region pairs, raw PCSs: humans = 1.52 [0.11, 3.02], macaques = 1.33 [0.33, 2.44], mice = 1.430 [0.230, 2.560]; z-scored PCSs: humans = 2.40 [1.30, 3.40], macaques = 2.00 [1.00, 2.78], mice = 2.00 [1.10, 2.90]). The PCS and z-scored PCS distributions for the three main (non-anesthetized) datasets are reported below for completeness (extracts from Fig. 2 and Supplementary Fig. 8):

Moreover, we performed additional statistical analyses to explicitly assess rank differences between the three species using Mann-Whitney U tests between group-average PCSs. When considering all the datasets (i.e., the three main datasets including non-anesthetized subjects, and the replication datasets), the gap in parallel communication mechanisms between humans and other mammalian species (macaques and mice) was strongly significant, while the differences between macaques and mice were minor (new Supplementary Fig. 2, reported below). We observed that human group-average PCSs are different (larger) than macaques and mice PCSs, independently from anesthesia (average effect size = 0.28).

Supplementary Figure 2 Parallel communication gap between humans and other mammalian species. Results from pairwise statistical comparisons between group-average parallel communication scores (PCSs) of humans, macaques and mice (all main and replication datasets included). Main (non-anesthetized) and replication (anesthetized) datasets are reported on rows and columns; an asterisk next to the dataset name indicates that animals were anesthetized. The colorbar represents group-difference effect size, computed as the difference of the medians normalized by the pooled median absolute deviation of the two groups. Violet tones indicate that the row dataset has larger PCSs than the column dataset; orange tones indicate that the row dataset has smaller PCSs than the column dataset. White asterisks in the matrix indicate p -values $< .05/15$ (Mann-Whitney U tests, Bonferroni corrected).

Besides group-average distributions, we investigated individual-level parallel communication scores. To this end, we assessed for each experimental subject the average PCS within unimodal brain systems, within transmodal brain systems, and between unimodal and transmodal brain systems ('cross-modal' connections) (Fig. 3c). In new analyses, we compared these individual-level scores between every pair of datasets, including both main (non-anesthetized) and replication datasets. The cross-species gap between human vs macaques and mice remained strongly significant at the individual subject level, with effects localized in cross-modal connections independently from animals' anesthesia (new Supplementary Fig. 5, reported below). This cross-species gap could not be explained by the number of anatomical connections, the brain spatial embedding, nor the resting-state functional connectivity alone (see detailed answer to next comment).

Supplementary Figure 4. Individual parallel communication scores across species and brain systems. Average PCSs between unimodal systems (top row), between transmodal systems (middle row), and interconnecting unimodal and transmodal systems (‘cross-modal’ region pairs; bottom row) for individual experimental subjects of the three mammalian species (humans: blue; macaques: green; mice: brick red), and for all main and replication datasets. Statistically significant pairwise comparisons are indicated with horizontal black lines (Mann-Whitney U tests; $p < .05 / 15$, Bonferroni-corrected multiple comparisons). Datasets including anesthetized animals are indicated with a gray-shaded rectangle and an asterisk. In the box plots, each dot represents an individual; vertical bars indicate mean \pm standard deviation; notch bars indicate median and 1st-3rd quartiles; shaded areas indicate 1st-99th percentiles.

Taken together, our group-level and individual-level results indicate the presence of a cross-species gap between humans and other mammalian animals, with human brain communication tailored towards parallel information transmission and other mammalian brains tailored towards selective information transmission. However, we do acknowledge that the concept of “evolutionary gradient” could be too strong for the reported analyses. We have therefore reworded the term “evolutionary gradient” with the more appropriate “cross-species gap”

between humans and other mammalian species in several parts of the manuscript. The new statistical analyses have been added to the manuscript.

To better understand the meaning of the parallel transmission pathway characteristic, please compare with other comparable approaches to connectivity, such as standard functional connectivity measures, number of structural connections (i.e., pretending that all structural connections are used for transmission). An interesting question is also how this differs for different brain areas as shown in Figure 3. Are there possible relationships to current approaches of assessing functional brain architecture and evolution?

We thank the Reviewer for this comment, which brought us to perform additional analyses. First, we investigated the relationship between regional parallel communication scores (as illustrated in Figure 3a) and (i) the number of structural connections (structural degree), (ii) the average length of structural connections, and (iii) the overall functional connectivity (functional strength) of individual brain regions. The relationship between PCS and brain connectivity features were assessed with a multiple regression model including the PCS as dependent variable, and the species and the three nodal connectivity measures as independent variables. There was a significant effect of the species, confirming that PCSs in macaques and mice are smaller than in humans. We found that all three connectivity measures explained significant variance of parallel communication scores (Bonferroni-corrected p -values $< 10^{-7}$; new Supplementary Table 2). The proportion of explained PCS variance (model's R -squared) was 0.42. In line with previous literature, these results indicate that multiple aspects of the connectome architecture shape (but do not completely determine) communication patterns, including the number of anatomical connections, the brain spatial embedding, and the resting-state functional coupling.

Second, mirroring Fig. 3c, we determined unimodal, transmodal and cross-modal connectome features for each species. We computed (i) the density of structural connections, (ii) the average length of structural connections, and (iii) the individual-subject average functional connectivity values between unimodal systems, between transmodal systems, and between unimodal and transmodal systems (*cross-modal connections*). The observed distributions (new Supplementary Figure 21) indicate that the cross-species gap, with strong parallel communication streams between unimodal and transmodal ('cross-modal') areas in humans (Fig. 3c), cannot be explained by structural and functional connectivity features alone.

We have included these new analyses in the updated manuscript, and we report below the new supplementary table and figure for completeness:

	Beta	SE	t statistic	p-value
Intercept	2.27	0.26	8.71	$4.7 \cdot 10^{-16}$
Species (macaque)	-3.40	0.35	-9.75	$3.7 \cdot 10^{-19}$

Species (mouse)	-4.31	0.59	-7.24	$5.9 \cdot 10^{-12}$
SC - number of connections (degree)	0.57	0.06	0.11	$3.3 \cdot 10^{-17}$
SC - average connection length	-0.68	0.19	-3.53	$4.9 \cdot 10^{-4}$
FC - MI sum	-1.14	0.17	-8.01	$4.7 \cdot 10^{-14}$

Supplementary Table 2. Relationship between parallel communication and connectome architecture. Multiple regression model including group-average nodal parallel communication scores (PCSs, as depicted in Fig. 3a) as dependent variable, and the species (human, macaque and mouse encoded as two linearly independent regressors) and three connectome nodal features (the number of structural connections (structural degree), the average length of structural connections, and the sum of functional connectivity weights) as independent variables (model F -statistic = 34.70, $p < 10^{-15}$; R -squared = 0.42). All variables were z-scored. The length of structural connections was quantified as Euclidean distance between regions' centroids; functional connectivity weights were quantified as group-average mutual information between fMRI time series. SC=structural connectivity; FC=functional connectivity; MI=mutual information; Beta=estimated coefficient; SE=standard error of estimated coefficient.

Supplementary Figure 21. Structural and functional connectivity features across unimodal and transmodal systems. Density of structural connections (a), average length of structural connections (quantified as Euclidean distance between regions' centroids) (b), and average functional connectivity in individual subjects (quantified as mutual information between regions' fMRI time series) (c) between unimodal systems, between transmodal systems, and between unimodal and transmodal systems (cross-modal communication). In (b) each dot represents a structural connection; in (c) each dot represents a subject, similarly to Fig. 3c. In the box plots, vertical bars indicate mean \pm standard deviation; notch bars indicate median and 1st-3rd quartiles; shaded areas indicate 1st-99th percentiles. Cross-species characteristics of parallel communication patterns, with strong parallel communication streams between unimodal and transmodal ('cross-modal') areas in humans (Fig. 3c), are not trivially explained by the number of anatomical connections, brain spatial embedding, and resting-state functional coupling alone.

Several minor points:

line 153: "at most" is not an assumption but a necessity if I understand correctly. Please clarify

or formulate precisely.

The Reviewer is right. Under a relay information transmission model, the statement indicates a necessity and not an assumption. We have reformulated the text as follows:

“We aimed to formally investigate two general aspects of brain communication dynamics. First, due to the noisy nature of neural signaling and under a relay communication model, neural messages transmitted through the structural brain network can keep at most the same amount of information present at the source region.”

line 222: similar for 3 species but decaying right after ... what does that mean?

We apologize for the lack of clarity. We have reformulated the sentence as follows:

“Specifically, the communication density levels were comparable for the first and second shortest paths and decreased for longer paths in all the three species.”

line 371 ff the parallelism or also it's increase remained significant when compared to null hypothesis? is this independent from overall connectivity of a region? compare to count of connections from a region of all paths are counted as valid path. or is this equivalent to the time series shuffling please clarify

Please, see our answer to the above comment. Following the Reviewer's suggestion, we have assessed the contribution of the overall structural connectivity (i.e., the count of connections from a region), the overall functional connectivity (i.e., the sum of functional connectivity values from a region) and the spatial embedding (i.e., the average length of the connections from a region) to the parallel communication scores. Our results, which have been added to the manuscript, indicate that structural and functional connectome features contribute to shape but do not explain the parallel communication patterns observed in the three species (Supplementary Table 2 and Supplementary Fig. 21 reported above).

please check grammar, e.g., line 58: give -> gives

Thank you. We have corrected grammar errors throughout the manuscript.

line 62: what do you mean by “virtually subtend all aspects“? Please make clear statements. Do you mean “subtend almost all aspects“

We have reformulated the sentence as suggested by the Reviewer:

“Communication processes are at the foundation of the brain's computational capacities that subtend almost all aspects of behavior, from sensory perception and motor functions shared across mammalian species, to complex human functions including higher-level cognition.”

References

Avena-Koenigsberger, A., Misic, B., Hawkins, R.X.D., et al., 2017. Path ensembles and a tradeoff between communication efficiency and resilience in the human connectome. *Brain Structure and Function* 222:603-618

Avena-Koenigsberger, A., Misic, B., Sporns, O., 2018. Communication dynamics in complex brain networks. *Nat Rev Neurosci* 19: 17–33

Estrada, E., Hatani, H., 2008. Communicability in complex networks. *Physical Review E* 77:036111.

Goñi, J., van den Heuvel, M.P., Avena-Koenigsberger, A., Velez de Mendizabal, N., et al., 2014. Resting-brain functional connectivity predicted by analytic measures of network communication. *PNAS* 111, 833–838

Luppi, A.I., Craig, M.M., Pappas, I., et al., 2022. A synergistic core for human brain evolution and cognition. *Nature Neuroscience* 25:771-782

Seguin, C., Tian, Y., Zalesky, A., 2020. Network communication models improve the behavioral and functional predictive utility of the human structural connectome. *Network Neuroscience* 4, 980–1006

Seguin, C., Jedynak, M., David, O., et al., 2023. Communication dynamics in the human connectome shape the cortex-wide propagation of direct electrical stimulation. *Neuron* 111, 1391-1401.e5

Zhou, D., Lynn, C.W., Cui, Z., et al. 2023. Efficient coding in the economics of human brain connectomics. *Network Neuroscience* doi:10.1162/netn_a_00223.

Reviewer #3 (Remarks to the Author):

This paper aims to explore the information transmission mechanisms in mammalian brain networks. First, the information transmission in brain networks is modeled as (multi-hop) relay communications. Second, the data processing inequality (DPI) in information theory is used to find all the structural paths between two brain regions which can serve as a relay communication channel. More specifically, a structural path is labeled as a communication channel if the pairwise mutual information values do not increase along the unidirectional path; otherwise, it is not a communication channel. For example, consider a path $A \rightarrow B \rightarrow C \rightarrow D$ and let $I(X, Y)$ denote the mutual information between regions X and Y . If $I(A, B) \geq I(A, C) \geq I(A, D)$, then $A \rightarrow B \rightarrow C \rightarrow D$ is labeled as a communication channel, otherwise it is not a communication channel.

Based on this approach, the authors analyzed the brain networks for mice, macaques, and humans, and found that, on average, the number of parallel communication path progressively increase from mice to macaques to humans, and the relay communication/routing pattern varies across species and different individuals in the same species. These conclusions are consistent with our common sense: parallel communications are critical for advanced cognitive functions in the human brain, and different people may have different connectivity patterns in the brain network due to different experiences and trainings. The paper is well written and easy to follow.

We thank the Reviewer for the evaluation of our work and the issues raised in the comments. We sincerely appreciated the reasoning through practical examples and we followed them up in our point-by-point answers listed below.

This reviewer has the following comments:

1. A path that follows DPI may not be a communication channel.

First, let's assume that the relay model is correct. While it is true that a memoryless relay communication channel generally follows DPI, a path that follows DPI may not be a communication channel.

Let's look at an example. Before that we need to understand what a communication channel is. I believe that we all have the experience that, sometimes, our phone call is dropped because the signal fades, or our Zoom link does not work when the network speed is too low. This means that a communication is possible only when the connectivity between the source and destination is strong enough to support effective information exchange between them. Consider a structural path $A \rightarrow B \rightarrow C \rightarrow D$, even if there is only noise in the observed signal, we are likely to observe that $I(A, B) \geq I(A, C) \geq I(A, D)$ (with all the mutual information values been nonzero as the noise there may not be independent), simply because that distance between brain regions increases along the path. However, in this case, the path

follows DPI, but there is actually no information transmission going on.

We thank the Reviewer for this example and agree with the concept of communication channel. Please note that with our null model, which randomly shuffles functional signals on top of a fixed structural topology, we already accounted for some “physiological noise” in brain networks.

Nonetheless, inspired by the Reviewer’s Zoom example, we have performed additional analyses and added two new null models to our study. The first null model represents the case of spatially structured physiological noise and simulates the ‘Zoom call noise’ scenario. To this end, we randomly permuted the mutual information values of individual subjects while preserving their spatial autocorrelation with respect to the underlying distance-weighted structural connectivity matrix. The second null model represents the case of absent communication. To this end, we populated the network nodes with iid Gaussian noise with mean 0, variance 1, and the same number of time points as in experimental data (100 simulations). The additional null models are illustrated in the updated manuscript:

“A second null model was defined by populating the network nodes with iid Gaussian noise with mean 0, variance 1, and the same number of time points as in experimental data (100 simulations). This scenario corresponds to the case of absent communication.

A third null model was developed for mutual information (MI)-based functional connectomes, which preserves their spatial autocorrelation with respect to the underlying distance-weighted structural connectivity matrix. Specifically, similarly to the brainSMASH algorithm⁴¹, we first quantified the level of spatial autocorrelation in the original individual MI matrices with a variogram. This is the target variogram for the output null MI matrices. Then, the original individual MI matrix is randomly permuted. Spatial autocorrelation among the samples is reintroduced by smoothing the permuted map with a distance-dependent kernel (similarly to BrainSMASH, here we employed an exponentially decaying gaussian kernel). Afterwards, the smoothed MI matrix’s variogram is computed and then regressed onto the variogram for the original MI matrix. The goodness-of-fit is quantified by computing the sum of squared error (SSE) in the null vs. original variogram fit. Finally, for every individual, we picked the null MI matrix with the best goodness-of-fit over 100 permutation runs.”

We assessed the number of DPI-respecting k -shortest paths in both scenarios of spatially structured physiological noise and Gaussian noise. The percentage of structural paths respecting the DPI in individual brain networks was significantly higher than in the null models, for all investigated species (Supplementary Fig. 19, 20 a). Moreover, the distribution of group-average PCSs significantly differed from null models (Supplementary Fig. 19, 20 b). Finally, we investigated whether structured and Gaussian noise may engender “communication” patterns similar to what observed in human subjects, i.e., larger parallel communication between unimodal and transmodal areas (“cross-modal” channels) compared to communication within unimodal or transmodal systems, as assessed with the DPI. We found that structured and Gaussian noise cannot explain system-dependent communication patterns (Supplementary Fig. 19, 20 c).

Note that, for all null models, we tested the DPI on a fixed structural topology—the real structural brain network weighted by the Euclidean distance between region pairs. In this way we account not only for the presence of (structured or unstructured) noisy signals at the network nodes, but also for the influence of the network topology (and brain spatial embedding), which is intrinsic to our multimodal framework. The influence of the network topology is evident in the fact that, even in case of Gaussian noise, the percentage of DPI-respecting structural paths is not zero (gray-dot distributions in Supplementary Figure 19a). As expected, the PCSs were larger in case of spatially structured physiological noise compared to Gaussian noise, although significantly smaller than real data (compare Supplementary Fig. 19 c and 20 c), .

Our analyses, which test functional brain data against three distinct null models, show that the amount and location of DPI-respecting k -shortest paths in the brain networks of different mammalian species are meaningful and cannot be explained by structured noise and spatial embedding alone.

We report the new analyses including the two additional null models in the updated manuscript. The new Supplementary figures are reported below for completeness:

Supplementary Figure 19. Comparison between real data and iid Gaussian noise null model. Left: drawings of human, macaque, and mouse brains; each row in the figure

corresponds to one species. **(a)** Box plots representing the percentage of short paths in individual brain networks used for relayed communication (i.e., respecting the DPI). Each colored dot represents an individual. Gray dots represent null distributions obtained by plugging in iid Gaussian noise at the nodes of the brain network; the brain network structural topology was unchanged (Online Methods). Circles and vertical bars indicate mean \pm one standard deviation across individuals and randomizations. Paths are grouped according to the 1st up to the 5th shortest path between region pairs. **(b)** Histograms of group-average PCSs obtained from real data (blue) and Gaussian null model (pink). For all the species, the real PCS histograms significantly differ from the null ones, indicating that parallel communication levels are not the trivial byproduct of the structural network topology (weighted by the Euclidean distance between region pairs) and brain spatial embedding. Yet, the effect of the structural network topology is evident in the fact that the median PCSs in case of Gaussian noise are larger than zero. **(c)** Average PCSs between unimodal systems, between transmodal systems, and between unimodal and transmodal systems (cross-modal communication) for individual experimental subjects. In the box plots, each dot represents a subject individual; vertical bars indicate mean \pm standard deviation; notch bars indicate median and 1st-3rd quartiles; shaded areas indicate 1st-99th percentiles. Values from the Gaussian null model are reported in the same panels as gray box plots. For all species and brain systems, the real individual PCS significantly differ from the null ones, indicating that the structural network topology cannot explain system-dependent spatial communication patterns.

Supplementary Figure 20. Comparison between real data and spatially structured physiological noise null model. Left: drawings of human, macaque, and mouse brains; each row in the figure corresponds to one species. **(a)** Box plots representing the percentage of short paths in individual brain networks used for relayed communication (i.e., respecting the DPI). Each colored dot represents an individual. Gray dots represent null distributions obtained by shuffling individual mutual information values while preserving network-level autocorrelation patterns (i.e., while preserving the mutual information variogram); the brain network structural topology was unchanged (Online Methods). Circles and vertical bars indicate mean \pm one standard deviation across individuals and randomizations. Paths are grouped according to the 1st up to the 5th shortest path between region pairs. **(b)** Histograms of group-average PCSs obtained from real data (blue) and spatially structured noise (pink). For all the species, the real PCS histograms significantly differ from the null ones, indicating that parallel communication levels are not the trivial byproduct of the network-level autocorrelation present in functional brain data. **(c)** Average PCSs between unimodal systems, between transmodal systems, and between unimodal and transmodal systems (cross-modal communication) for individual experimental subjects. In the box plots, each dot represents a subject individual; vertical bars indicate mean \pm standard deviation; notch bars indicate median and 1st-3rd quartiles; shaded areas indicate 1st-99th percentiles. Values from the spatially structured null model are reported in the same panels as gray box plots. For all species and brain systems, the real individual PCS significantly differ from the null ones, indicating that the network-level spatial autocorrelation of functional signals cannot explain system-dependent spatial communication patterns.

The Results and Discussion now reads:

“Parallel communication scores and unimodal-transmodal variations in parallel communication were not explained by the different structural architectures and brain spatial embedding of the investigated species, nor by the spatial autocorrelation of the mutual information values with respect to the underlying distance-weighted structural connectivity matrix. To test these aspects, we constructed two additional null models. In the first one we populated the network nodes with iid Gaussian noise with mean 0, variance 1, and the same number of time points as in experimental data. This scenario represents the case of absent communication between brain regions interconnected through the real structural connectome. In the second model we randomly permuted the mutual information values of individual subjects while preserving their spatial autocorrelation with respect to the underlying distance-weighted structural connectivity matrix, similarly to the method proposed in⁴¹ (see Online Methods). Parallel communication densities and PCS scores (group-average and individual-level PCSs of unimodal, transmodal and cross-modal communication streams) were larger than expected based on the connectome spatial embedding and spatial autocorrelation of the mutual information values alone (Supplementary Fig. 19, 20).”

“Relative cross-species differences of parallel communication were unchanged when contrasting data with respect to different species-specific null models which preserve multivariate fMRI statistics, spatial autocorrelation of mutual information with respect to the underlying distance-weighted structural connectivity matrix, and/or the structural connectome architecture.”

Second, since mutual information reflects the mutual dependence between the brain regions and is nondirectional, a path that follows DPI may not be a communication channel.

Consider an example $A \leftarrow B \rightarrow C$ (just as in the case that B is a friend for both A and C, but A and C do not really know each other), we are likely to have $I(A,B) \geq I(A,C)$, and then the proposed approach will assume that the $A \rightarrow B \rightarrow C$ is a communication channel but this is not true.

The Reviewer is correct. Since we use a measure of undirected information exchange between regions pairs, our framework cannot distinguish the cases mentioned above ($A \leftarrow B \rightarrow C$, $A \rightarrow B \rightarrow C$, etc.). As also pointed out by the Reviewer below, different measures of directed information exchange have been proposed at the theoretical level. Nonetheless, the estimation of directed information in noisy experimental data at the low spatial and temporal resolutions achieved with diffusion and functional MRI is challenging and relies on strong assumptions on data distribution. Diffusion tractography is affected by several limitations, including uncertainty in diffusion direction estimation, hindering the reconstruction of axonal connections at the mesoscopic scale (Maier-Hein et al., 2017). Although single axonal projections are intrinsically directed, the whole-brain structural connectome reconstructed from diffusion tractography is undirected. Functional MRI relies on the cerebrovascular coupling and is sensitive to slow

changes of blood oxygenated and deoxygenated hemoglobin in gray matter regions. The hemodynamic response to an ideal isolated neural event lags several seconds from the event's onset and lasts up to 20 seconds (Lindquist et al., 2009). These factors partially hinder the assessment of directed information from fMRI experimental data. In a first information theory model of relay communication in multimodal MRI brain networks, we therefore approximate information exchange with a simple undirected measure (mutual information) and model undirected relay communication patterns of the form $A \leftrightarrow B \leftrightarrow C$ through the DPI construct.

However, this is only one of the possible communication scenarios that could happen in a (resting-state) brain network. In fact, there is a large landscape of possible communication scenarios, such as loops and feedback, that we are excluding with the DPI-based communication model (please, see also our answer to the next comment). Yet, in our view the DPI-based model represents a first starting point in applying information theory constructs to noisy and complex data derived from multimodal brain MRI, which lack any type of ground truth in terms of senders, receivers, and communication scenarios. Specifically, the model attempts to answer the following research question (following the reviewer's notation): *assuming that A is transmitting information to C, is this exchange of information compatible with a relay communication channel (in the DPI-sense)?* Backed up by previous literature (Avena-Koenigsberger et al., 2019), we looked at the k -shortest paths that are considered (based on the DPI) *possible* channels of relay information transmission. We acknowledge that this assumption is a strong one, and by no means we aimed at explaining the entire spectrum of communication in brain networks within this single paper.

We elaborate more on these aspects in the updated Results and Discussion, which now read:

“We aimed to investigate two general aspects of brain communication dynamics. First, due to the noisy nature of neural signaling and under a relay communication model, neural messages transmitted through the structural brain network can keep at most the same amount of information present at the source region^{16,35}. This holds true for many communication systems where the information content tends to decay as one moves away from the information source²⁵. Second, in an information transmission process, messages are typically relayed through a set of statistically independent steps⁴⁴; i.e., neural messages do not contain memory of the transmission process itself and communication happens in a Markovian fashion³⁷. These two dimensions –information decay and memoryless transmission– are formally summarized by a fundamental principle of information theory, the data processing inequality (DPI)³⁶, which we here apply to cross-species structural data and fMRI recordings (Online Methods). Our model answers the following research question: assuming that a source brain region is transmitting information to a destination region, is this exchange of information compatible with a relay communication channel (in the DPI-sense)?”

“The framework proposed in the present work only models Markovian information transmission and does not inform us about other complementary brain communication mechanisms. Moreover, we used an undirected measure of information exchange between regions pairs, the

mutual information, which is well adapted to the spatial and temporal resolution of fMRI recordings. Therefore, our framework cannot resolve ‘star’ relay information motifs with a central node being the source of information in the communication process, which is one of its limitations. Different measures of directed information exchange have been proposed in literature^{60,61} and should be explored in future work in relation to the structural connectome architecture and communication mechanisms. By no means the proposed model aims at explaining the entire spectrum of communication mechanisms in brain networks.”

2. A path that does not follow DPI may be a communication channel.

Let’s look at an example. Consider a path $X \rightarrow Y \rightarrow Z$. Assuming X and Z are students, and Y is a teacher. X reports to Y about what he is doing; Y then asks Z to contact X and does the same thing X is doing; Z contacts X and they exchange information. Recall that physical meaning of mutual information is the amount of information successfully exchanged between the two regions. In this example, we are going to see that $I(X,Z) \geq I(X,Y)$, it does not follow DPI; however, $X \rightarrow Y \rightarrow Z$ surely is a communication path.

Coming back to the brain, if each brain region just performs information relay without, then the communication will not be meaningful and it will be impossible for us to perform any cognitive thinking. The brain might be more likely to follow a receiving-processing-transmitting model rather than a simple relay communication model.

As aforementioned, we agree with the Reviewer that the DPI-based relay information transmission represents a simplistic model of brain communication and excludes multiple scenarios such as higher-order interactions (in the Reviewer’s example, Z and X need the mediation of Y to communicate) and processing operations (Y interprets the message received from X and transmit a new message to Z). All these scenarios represent plausible communication *models* for large-scale brain networks. Previous work showing a strong spatial heterogeneity of structure-function coupling across the cortical mantle, with low coupling in transmodal regions, has already suggested that brain communication mechanisms may be multiplexed, with multiple protocols operating in parallel (Vázquez-Rodríguez et al., 2019). Yet, communication models represent approximations of brain functioning at the large-scale level and remain far from matching actual communication mechanisms at the neuronal level.

In this first work we only model relay (Markovian) information exchanges. Despite the model’s simplicity, the results presented here show that undirected Markovian patterns are present in different mammalian brain networks beyond what can be explained by spatially structured physiological noise, structural embedding, and multivariate statistical properties of fMRI time series; they highlight interesting cross-species difference and are specific to individual experimental subjects.

We note that simpler models of brain communication, such as functional connectivity (Pearson’s

correlation) between region pairs, have proven powerful in multiple neuroscience branches, from explaining inter-individual cognitive and behavioral characteristics (Dadi et al., 2019; Finn et al., 2015) to clinical applications (Amico et al., 2017). Functional connectivity (FC) does not explain how the brain works at the neuronal level, and yet it is an effective (linear) model for brain signals, while not capturing all the subtleties inherent to neural signaling. Similarly to FC, our DPI framework is, in the first place, a model to explain the observed BOLD signals. Yet, by applying information theoretical constructs that help bridging structure and function together, and by proposing a new (although simple) model of Markovian information exchange through structural pathways, the framework represents a step forward compared to the FC model.

Despite these considerations, we are aware that future work should include and characterize also the aforementioned higher-order communication mechanisms and information processing operations. This is now better highlighted in the Discussion, which now reads:

“Polysynaptic memoryless information transmission is a simple model of communication. There is no reason to assume that macroscale neural communication is limited to such a particular form. Brain network hierarchies may confer neural signals a memory of the regions previously visited along a path, thus modifying neural communication pathways in a context-dependent manner⁵⁵. This process would result in non-Markovian communication regimes. The brain may also be modeled through complex multi-object interactions not attributable to information transmission alone, such as synergistic or modulatory behaviors between multiple brain regions; feedback loops; local transformation (non-linear processing) of information^{19,40,58,59}. Biologically, these more complex communication patterns may shape important features of the mammalian brain, such as cortical temporal hierarchies^{57,58} or receptive time windows for attentional processes⁵⁹, and are worthy to investigate in future studies. Previous work showing a strong spatial heterogeneity of structure-function coupling across the cortical mantle has suggested that brain communication mechanisms may be multiplexed, with multiple protocols operating in parallel⁴⁶. The framework proposed in the present work only models Markovian information transmission and does not inform us about other complementary brain communication mechanisms. ... By no means the proposed model aims at explaining the entire spectrum of communication mechanisms in brain networks. As such, absence of relay communication (i.e., violation of the data processing inequality) may indicate absence of any communication between those particular brain regions, or communication through more complex information encoding mechanisms.”

3. Unidirectional information transmission, in general, is measured using directed information rather than mutual information. Mutual information and directed information are equal only when there is no feedback in the channel. For directed information, the authors may want to refer to the following paper:

J. L. Massey and P. C. Massey, "Conservation of mutual and directed information," Proceedings. International Symposium on Information Theory, 2005. ISIT 2005., 2005, pp. 157-158, doi: 10.1109/ISIT.2005.1523313.

Thanks for the suggestion. Indeed, one future avenue could be to merge directed information communication matrices with PCS to gain more detailed insight on the different communication scenarios in brain networks. It will be particularly interesting to test directed and higher-order models on data with rich spatio-temporal information, such as intracranial EEG—able to record electrical activity at millisecond resolution and track causal propagation of interventional stimulations; see Seguin et al., 2023 for an example) — and calcium imaging — able to track spatio-temporal signaling at the single cells' level.

We now better discuss possible avenues related to directed information quantification:

“Different measures of directed information exchange have been proposed in literature^{60,61} and should be explored in future work in relation to the structural connectome architecture and communication mechanisms. It will be particularly interesting to test directed and more complex communication models on data with rich spatio-temporal information, such as intracranial EEG (see ⁵¹ for an example) and calcium imaging.”

Overall, the topic is very interesting, and I appreciate the numerical analysis the authors have carried out. However, unfortunately, I am very sorry to say that using the DPI approach for communication path detection in brain networks is not technically sound.

Please, see our replies to the points above.

In this work we propose to the community a novel way of bridging brain structural and functional information, which tries to go beyond pairwise functional connectivity and structure-function correlations by modeling relay communication patterns via information-theoretical tools.

Our approach is empirical in spirit and, as all models, presents several limitations which are now better highlighted in the Discussion and in the description of the framework. Future work should refine, relax and extend this first study; for instance, by using directed information to filter and select “pure” relay pathways from hybrid ones (as the $A \leftarrow B \rightarrow C$ example illustrated by the Reviewer); or by using higher-order information theoretical metrics (such as O-Information, see Varley et al., 2023 for an example on brain networks) to decompose the contribution of communication and synergistic effects along the different pathways.

The DPI-based model is restricted to relay communication patterns and discards several possible communication scenarios. Yet, it delivers a novel perspective on brain structure and function and highlights communication patterns that cannot be explained by Gaussian noise, structured physiological noise, or spatial embedding. From a neuroscientific perspective, these communication patterns are meaningful in that they highlight novel and well interpretable cross-species differences, brain systems' heterogeneity and individual-specific traits.

I believe that analyzing brain network using information theory is perhaps the right direction, but this particular approach should be rethought.

We thank the Reviewer for the insightful comments, which helped us rethinking and better clarifying assumptions and limitations of our model, and which encouraged us to introduce additional null scenarios that have strengthened the interpretation of our results. By no means we aimed at explaining the entire spectrum of communication in brain networks within this single paper.

We hope that our previous clarifications convinced the Reviewer that the proposed approach, interpreted at the right level, brings new insights to the field of network neuroscience.

References

Amico 2017 Mapping functional connectome traits of consciousness, *NeuroImage*

Avena-Koenigsberger, A., Misic, B., Sporns, O., 2018. Communication dynamics in complex brain networks. *Nat Rev Neurosci* 19: 17–33

Dadi, K., Rahim, M., Abraham, A., Chyzyk, D., Milham, M., Thirion, B., Varoquaux, G., 2019. Benchmarking functional connectome-based predictive models for resting-state fMRI. *NeuroImage* 192: 115-134

Finn, E., Shen, X., Scheinost, D. et al., 2015. Functional connectome fingerprinting: identifying individuals using patterns of brain connectivity. *Nat Neurosci* 18: 1664–1671

Lindquist, M.A., Loh, J.M., Atlas, L.Y., Wager, T.D., 2009. Modeling the hemodynamic response function in fMRI: Efficiency, bias and mis-modeling. *NeuroImage* 45(1): S187-S198

Maier-Hein, K.H., Neher, P.F., Houde, J.C., et al., 2017. The challenge of mapping the human connectome based on diffusion tractography. *Nature Communications* 8: 1349

Seguin, C., Jedynak, M., David, O., et al., 2023. Communication dynamics in the human connectome shape the cortex-wide propagation of direct electrical stimulation. *Neuron*:01-027

Varley, T.F., Pope, M., Faskowitz, J., Sporns, O., 2023. Multivariate information theory uncovers synergistic subsystems of the human cerebral cortex. *Communications Biology* 6: 451

Vázquez-Rodríguez, B., Suárez, L.E., Markello, R.D., Shafiei, G., Paquola, C., Hagmann, P., Heuvel, M.P. van den, Bernhardt, B.C., Spreng, R.N., Misic, B., 2019. Gradients of structure–function tethering across neocortex. *Proc. Natl. Acad. Sci.* 116: 21219–21227

Reviewer #1 (Remarks to the Author):

I appreciate the effort the authors have put into their revisions.

However, I still have several concerns/questions.

1. The first is that their result, namely the gap in PCS across species, is only significant for humans v. others, while not significant for monkeys v. mice. This seems to be at odds with some of the fundamental claims of their paper. Are monkeys and mice really not that dissimilar? Do monkeys/mice belong to the same part of the 'evolutionary' tree? The ethological behavior of these animals is remarkably different, and should not be overlooked.

2. I am still skeptical of the binarization of the SC weights, with subsequent modification of these weights according to the Euclidean distance. Subsequent measures, such as PCS were used on this matrix. This seems like a strong assumption, given the fact that the brain already is biased towards having local connections; it's not clear to me that weighting by euclidean distance is required further. I would be more confident of this result if the PCS results are able to be maintained without modification of the weights using Euclidean distance (e.g., either using the binarized matrix or the weights derived from the unfiltered SC matrix). Or, if one were to parameterize an all-to-all connectivity matrix with euclidean distance, do the results emerge? In the latter case, this would suggest that PCS has nothing to do with SC, but instead euclidean distance.

3. Regarding the evaluation and comparison of relay communication patterns that are unique to individuals, the authors suggest a potential workaround is to cut the resting state time series in half, which has the benefit of removing scanner and acquisition noise. I don't understand how this removes scanner and acquisition noise – doesn't this do the opposite, i.e., add scanner and acquisition noise as a confound?

4. Regarding the use of evolutionary gap when comparing humans, monkeys and mice. The authors mention that they have "smoothed out the claims of evolutionary gradient" throughout the text. However, when reviewing the manuscript, the concept still appears to be pervasive. A few examples:

Abstract, line 46: "By applying a novel approach to measure information transmission in mouse, macaque and human brains, we found an evolutionary gap from selective information processing in phylogenetically older species..."

Introduction, line 129: "We report a strong evolutionary gap in the brain communication dynamics of mammals, with predominant selective information routing in phylogenetically older species such as mice and macaques, morphing into more complex communication patterns in human brains."

Reviewer #3 (Remarks to the Author):

The authors responded to my comments by adding more simulation results and identifying the limitations or simplicity of the proposed approach. I appreciate the authors effort in that.

As the authors indicated: "The framework models memoryless information transmission in brain networks by introducing an empirical way to assess deviations from Markovity through the data processing inequality", "The framework proposed in the present work only models Markovian information transmission and does not inform us about other complementary brain communication mechanisms."

Along this line, please note that nonzero mutual information between two brain regions does not necessarily mean there is actual information transmission between them. The frequently appeared

phrase "communication pathway" is therefore very strong and may be misleading since "communications" means "information transmission" either in one direction or both directions.

I agree with the author that one paper cannot solve all the problems on this complex topic and the empirical observation in the decreasing of mutual information (MI) along with the physical distance of the nodes is intuitively correct and aligns well with our common sense. From an objective perspective, it is more appropriate to report this decreasing pattern in the "information related link" of MI rather than claiming the link be a "communication path". I understand that this may sound less cool, however, it is more objective and safe.

That being said, I support the publication of the work if this drawback can be fixed.

Reviewer #4 (Remarks to the Author):

Griffa et al. used a combination of anatomical and functional scans together with an information-theory analysis to compare the brains of three species and claim that the human brain has more parallel computing. The combination of functional and anatomical data across species is unique and interesting. I was asked to assess the answers they gave reviewer 2 which I think were satisfactory. However, I have several major concerns that echo the concerns of reviewer 1.

In general, I think that comparing brains of different sizes that were scanned on different machines with different SNR is almost impossible. The differences found between the brains are very probably a result of scanning differences. The authors perform several controls on the fMRI data, but I would be very concerned about the DWI data. Brains of different sizes scanned in different machines would result in different fiber depth which could easily generate such results. This is especially worrying as the smaller brains (which are more difficult to reconstruct) appear different from the larger brain. Larger brains enjoy better SNR and will allow better fiber tracking and reconstruction. Such an artifact is also suggested by the distribution of node-hops in the k-strongest connections presented by the authors (SI 1B) in response to a reviewer's comment where humans seem to have slightly more longer connections.

The best solution for this would be to scan all animals in the same machine, using the same resolution of scanning. Another, less desirable, option is to degrade the resolution of the larger brains so that it is the same as the smaller brain (and perhaps to equalize the noise) and to reconstruct the network again. Without some very convincing control for network reconstruction across species, my guess would be that it is driving the results.

I am also not sure about the information theory approach. In general, I like the idea of measuring multiple connections between nodes and going beyond the mean short path. However, I have a few questions about the method.

- 1) It is my understanding that the author did not use the strength of the mutual information between nodes – two nodes might have a very high or a very low mutual information, but they account them as connected as long they follow the decaying information rule.
- 2) The authors remove non markovian connections because the mutual information does not drop – but doesn't this mean that these nodes are actually more strongly connected (their MI is higher) ? Aren't they actually looking at the less connected nodes ?
- 3) As already mentioned by other reviewers the criterion of decreasing MI seems somewhat arbitrary. I could suggest a different criterion – find the nodes with the highest MI, and estimate the connectivity of each node based on it.

Minor comments:

There is no such thing as older species. All mammals existing today evolved over the same time period. The human brain could in theory be older than the mouse brain – unless you have evidence proving otherwise.

Response to Reviewers

Reviewer #1

I appreciate the effort the authors have put into their revisions.

However, I still have several concerns/questions.

1. The first is that their result, namely the gap in PCS across species, is only significant for humans v. others, while not significant for monkeys v. mice. This seems to be at odds with some of the fundamental claims of their paper. Are monkeys and mice really not that dissimilar? Do monkeys/mice belong to the same part of the 'evolutionary' tree? The ethological behavior of these animals is remarkably different, and should not be overlooked.

We agree with the Reviewer. In fact, thanks to the Reviewer's comments during the first round of evaluation, we have performed extensive analyses and better stated the main conclusion of this work in terms of a *gap* between humans and non-human mammals, rather than an *evolutionary gradient* from humans, to macaques, to mice.

We have now removed the concept of '*evolutionary*' gap from the manuscript, and replaced it with '*brain communication*' gap between humans and non-human mammals (see also our answer to Comment 4). We believe that the updated wording appropriately reflects the reported findings.

We acknowledge that macaques and mice do not occupy the same part of the mammalian evolutionary tree and that future work, eventually including more advanced models of brain communication, should delve deeper into this topic. We now elaborate more on this aspect in the relevant paragraph of the Discussion. Yet, we think that it is noteworthy to report the robust, replicable, and non-trivial finding of PCS differences between humans and non-human mammals. The Discussion now reads:

"... Parallel communication could therefore represent a more complex form of information transmission beyond hierarchical processing, which might support integration of perceptual modalities into more complex textures of cognition. Yet, it is important to remark that macaques and mice do not occupy the same part of the mammalian evolutionary tree and that future work, eventually including advanced models of brain communication beyond relay information transmission, should delve deeper into cross-species variations of communication patterns."

2. I am still skeptical of the binarization of the SC weights, with subsequent modification of these weights according to the Euclidean distance. Subsequent measures, such as PCS were used on this matrix. This seems like a strong assumption, given the fact that the brain already is biased towards having local connections; it's not clear to me that weighting by euclidean distance is required further. I would be more confident of this result if the PCS results are able to

be maintained without modification of the weights using Euclidean distance (e.g., either using the binarized matrix or the weights derived from the unfiltered SC matrix). Or, if one were to parameterize an all-to-all connectivity matrix with euclidean distance, do the results emerge? In the latter case, this would suggest that PCS has nothing to do with SC, but instead euclidean distance.

We understand the concern of the Reviewer. The decision of weighting the structural connections by the Euclidean distance between region pairs was driven by the heterogeneity of methods for SC estimation across species (tractography for humans; tractography and tract tracing for macaques; tract tracing for mice). In this respect, the Euclidean distance can be considered a common metric that can be applied to the structural connectomes of different species, obtained in different ways. Moreover, the Euclidean distance is a less rudimental connectome-weighting strategy than the binary information alone. The decision of weighting the structural connections by the Euclidean distance is also coherent with the information theory framework. As also mentioned by Reviewer 3, decreasing mutual information along with the physical distance of the nodes is intuitively correct and aligns well with our common sense.

Nevertheless, we followed the Reviewer's suggestion and tested how PCS matrices would appear when computing the k-shortest paths on the binary SC matrices (i.e., with no Euclidean distance weighting) of the three different species. We observed that the reported communication gap between humans and non-human mammals is respected also in this case (see figure below). We are therefore confident that our main results hold independently from the Euclidean distance weighting of the structural connectomes.

Left: Group-average parallel communication score (*PCS*) matrices representing *PCS*s between every pair of brain regions, averaged across individuals. The *PCS* scores were computed based on the k-shortest paths extracted from binary structural connectomes. **Right:** the histograms of the average *PCS* scores across region pairs highlight a brain communication gap between humans (with higher *PCS*s and presence of a skewed tail indicating parallel communication) and non-human mammals (with lower *PCS*s and mainly selective information processing). Median [5-, 95-percentile] *PCS* values for each species are reported atop each histogram. VIS=visual; SM=somatomotor; DA=dorsal attention; VA=ventral attention; L=limbic; CN=control; DMN=default mode; BR=barrel; AD=auditory networks.

3. Regarding the evaluation and comparison of relay communication patterns that are unique to individuals, the authors suggest a potential workaround is to cut the resting state time series in half, which has the benefit of removing scanner and acquisition noise. I don't understand how this removes scanner and acquisition noise – doesn't this do the opposite, i.e., add scanner and acquisition noise as a confound?

It is quite established that between-scanner or multi-site acquisitions and their subsequent analyses include the scanner-dependent variability that can mask true underlying changes in brain structure and function. Even when using identical (let alone “comparable”) imaging sequences and parameters, potential site-dependent differences might arise due to a range of physical variables, including field inhomogeneities, transmit and receive coil configurations, system stability, system maintenance, scanner drift over time, and many others (Friedman and Glover, 2006; Van Horn and Toga, 2009; Voyvodic, 2006). Determining and minimizing these unwanted site-dependent variations have become critical elements in the design of multi-site fMRI studies. Extensive work has also reported that individual identifiability is increased when looking at within-scanner fingerprinting (Amico and Goni, Scientific Reports 2018), as opposed to between-scanner or between-site fingerprinting (Bari et al., NeuroImage 2019). We have clarified this point in the relevant Discussion paragraph, that now reads:

*“This has the benefit of removing the scanner and acquisition noise, which is usually a major confound in connectome identification⁴⁹. It is quite established that between-scanner or multi-site acquisitions and their subsequent analyses include the scanner-dependent variability that can mask true underlying changes in brain structure and function. **In fact, it is known that even when using identical (let alone “comparable”) imaging sequences and parameters, potential site-dependent differences might arise due to a range of physical variables, including field inhomogeneities, transmit and receive coil configurations, system stability, system maintenance, scanner drift over time and many others (Friedman and Glover, 2006; Van Horn and Toga, 2009; Voyvodic, 2006).** However, it comes at the cost of looking at within-session fingerprinting, hence focusing more on the temporal stability aspect of the communication pattern, rather than on standard between-session identification. Nonetheless, it is noteworthy that humans could be better identified than animals solely on the basis of their (within-scanner) parallel communication profiles. Future studies should explore*

how the results change when considering multiple and longer sessions, whenever available, to estimate cross-species communication fingerprints.”

References

Friedman, L., Glover, G.H., The FBIRN Consortium, 2006. Reducing interscanner variability of activation in a multicenter fMRI study: controlling for signal-to-fluctuation-noise-ratio (SFNR) differences. *Neuroimage* 33, 471–481.

Van Horn, J.D., Toga, A.W., 2009. Multisite neuroimaging trials. *Curr. Opin. Neurol.* 22, 370–378

Voyvodic, J.T., 2006. Activation mapping as a percentage of local excitation: fMRI stability within scans, between scans and across field strengths. *Magn. Reson. Imag.* 24, 1249–1261.

Amico, E., Goni, J., 2018. The quest for identifiability in human functional connectomes. *Sci. Rep.* 8, 8254

4. Regarding the use of evolutionary gap when comparing humans, monkeys and mice. The authors mention that they have “smoothed out the claims of evolutionary gradient” throughout the text. However, when reviewing the manuscript, the concept still appears to be pervasive. A few examples:

Abstract, line 46: “By applying a novel approach to measure information transmission in mouse, macaque and human brains, we found an evolutionary gap from selective information processing in phylogenetically older species...”

Introduction, line 129: “We report a strong evolutionary gap in the brain communication dynamics of mammals, with predominant selective information routing in phylogenetically older species such as mice and macaques, morphing into more complex communication patterns in human brains.”

Please, see our answer to Comment 1. We have replaced ‘*evolutionary gap*’ with ‘*brain communication gap*’, leaving to the Discussion the interpretation (and limitations) of the results in terms of evolution. We believe that the updated wording appropriately reflects the reported findings.

The two sentences mentioned by the Reviewer now read:

“By applying a novel approach to assess information transmission in mouse, macaque and human brains, we found a brain communication gap from selective information processing in phylogenetically older species, where brain regions share information through single

polysynaptic pathways, to parallel information processing in humans, where regions communicate through multiple parallel pathways.”

“We report a brain communication gap between humans and non-human mammals, with predominant selective information routing in phylogenetically older species such as mice and macaques, morphing into more complex communication patterns in human brains.”

Reviewer #3

The authors responded to my comments by adding more simulation results and identifying the limitations or simplicity of the proposed approach. I appreciate the authors effort in that.

As the authors indicated: “The framework models memoryless information transmission in brain networks by introducing an empirical way to assess deviations from Markovity through the data processing inequality”, “The framework proposed in the present work only models Markovian information transmission and does not inform us about other complementary brain communication mechanisms.”

Along this line, please note that nonzero mutual information between two brain regions does not necessarily mean there is actual information transmission between them. The frequently appeared phrase “communication pathway” is therefore very strong and may be misleading since “communications” means “information transmission” either in one direction or both directions.

I agree with the author that one paper cannot solve all the problems on this complex topic and the empirical observation in the decreasing of mutual information (MI) along with the physical distance of the nodes is intuitively correct and aligns well with our common sense. From an objective perspective, it is more appropriate to report this decreasing pattern in the "information related link" of MI rather than claiming the link be a “communication path”. I understand that this may sound less cool, however, it is more objective and safe.

That being said, I support the publication of the work if this drawback can be fixed.

We would like to thank this Reviewer again for the insightful comments which helped us to significantly improve this work. Following the Reviewer’s suggestion, we have reworded ‘communication pathway’ as ‘information-related pathway’ in the updated version of the manuscript.

Reviewer #4

Griffa et al. used a combination of anatomical and functional scans together with an information-theory analysis to compare the brains of three species and claim that the human brain has more parallel computing. The combination of functional and anatomical data across species is unique and interesting. I was asked to assess the answers they gave reviewer 2 which I think were satisfactory. However, I have several major concerns that echo the concerns of reviewer 1.

We thank the Reviewer for the constructive comments which we address below.

In general, I think that comparing brains of different sizes that were scanned on different machines with different SNR is almost impossible. The differences found between the brains are very probably a result of scanning differences. The authors perform several controls on the fMRI data, but I would be very concerned about the DWI data. Brains of different sizes scanned in different machines would result in different fiber depth which could easily generate such results. This is especially worrying as the smaller brains (which are more difficult to reconstruct) appear different from the larger brain. Larger brains enjoy better SNR and will allow better fiber tracking and reconstruction. Such an artifact is also suggested by the distribution of node-hops in the k-strongest connections presented by the authors (SI 1B) in response to a reviewer's comment where humans seem to have slightly longer connections.

We agree with the Reviewer that this is indeed a crucial and fundamental issue for all studies comparing data from different species (see for example Luppi et al., *Nature Neuroscience* 2022 for a related study).

However, please note that in this study the structural connectomes were derived from different techniques in the three species: DWI for humans; an optimal combination of DWI and tract-tracing for macaques, nicely provided by The Virtual Brain project (Shen et al., *Scientific Data* 2019); and tract-tracing for mice, provided by the Allen Institute (Oh et al., *Nature* 2014). While the SNR is not directly comparable among the three species because of the heterogeneity of the techniques, the usage of tract-tracing arguably compensates for the difficulties of mapping structural connectivity in smaller brains. Although the structural connectivity matrices estimated from DWI and tract-tracing are only an approximation of the true underlying structural connectivity architecture, to our knowledge there is no reason to believe that tract-tracing is less accurate than DWI tractography. We apologize if the heterogeneity of methods for the estimation of structural connectivity matrices was not clear enough. This is now better stated early in the text (Results section):

“These structural paths are made of several hops from cortical region to neighborhood cortical region in the structural connectome. We note that the structural connectomes were estimated with different modalities in the three species (diffusion tractography and/or tract tracing) to cope with the difficulties in estimating diffusion-based connectomes in smaller brains (Online Methods).”

Figure SI 1B pictures the percentage of 1-hop, 2-hop, etc. structural paths among the k -shortest paths ($k = 1$ to 5) reconstructed for each species. We now explicitly compare the length (expressed as number of hops) of the k -shortest paths reconstructed for the different species. There was no statistically significant difference between the k -shortest paths length of humans and macaques (Mann-Whitney U test, $p = .98$). However, mice had on average longer k -shortest paths than both humans ($p < 10^{-3}$) and macaques ($p < 10^{-3}$). Humans therefore have slightly *shorter* paths than mice, but similar path length to macaques. Considering that our results show a parallel communication gap not only between humans and mice, but also between humans and macaques, it is unlikely that the k -shortest path lengths drive our results. We thank the Reviewer for having brought up this point, which is now addressed in the text (Results section):

“Despite the fact that white matter volume increases from mice, to macaques, to humans, which may increase structural paths’ count and thus opportunities for parallel processing, there was no meaningful cross-species difference in density, i.e., relative number of white matter connections (Supplementary Fig. 1; note that connectivity density and paths’ count depend on the methodological choices for white matter connectivity mapping). Moreover, individual PCS scores did not depend on structural connectivity density (Supplementary Fig. 12). There was no difference between human and macaque in the k -shortest path length as quantified by the number of hops along the structural paths (mean (standard deviation) over all region pairs, human: 2.43 (0.58); macaque: 2.43 (0.59); Mann-Whitney U test, $p = .98$). However, the mouse had on average slightly longer k -shortest paths (2.48 (0.60)) than both the human ($p < 10^{-3}$) and the macaque ($p < 10^{-3}$). Considering the gap in parallel communication between humans and non-human mammals (both macaques and mice), we can rule out that cross-species differences in parallel communication are driven by k -shortest path lengths.”

The best solution for this would be to scan all animals in the same machine, using the same resolution of scanning. Another, less desirable, option is to degrade the resolution of the larger brains so that it is the same as the smaller brain (and perhaps to equalize the noise) and to reconstruct the network again. Without some very convincing control for network reconstruction across species, my guess would be that it is driving the results.

Comparing brains of different species is indeed a challenging task. Even the same imaging machine would not optimally compensate for different brain size and other properties. In that sense, we find the approach of relying on the best-available technology and methodology for each species (DWI and/or tract-tracing) is the most reasonable approach.

Starting from state-of-the-art connectivity data, we decided to match the structural connectivity matrices of the three species in graph theory terms. All the three connectomes have a similar number of nodes and comparable graph density. Size and density are trivial drivers of more complex graph measures and should therefore be carefully controlled when comparing different brain graphs. Moreover, we note that this choice is coherent with the Reviewer idea of

downsampling the larger brains, in that one human region corresponds to a larger volume than one mouse region (because we approximately preserved the number of nodes across species). In our humble opinion, degrading the resolution of the human DWI data would not ease the comparison between humans and non-human mammals because of the heterogeneity of the structural connectivity estimation methods (see our answer above).

In line with the literature comparing brain connectivity features between different species (van den Heuvel, Bullmore and Sporns, *Trend in Cognitive Sciences* 2016; Luppi et al., *Nature Neuroscience* 2022), we performed several analyses to rule out possible confounding factors ('Robustness, sensitivity and replication analyses' in the Results section):

(1) results are not driven by structural network complexity as quantified by graph density and k -shortest path length (we again thank the Reviewer for bringing up this point);

(2) results are not explained by the three species' structural connectivity architectures alone, as tested by inputting i.i.d. Gaussian noise into the three structural connectomes;

(3) results are robust when considering several replication datasets, with animals scanned on different scanners and with different sequence parameters;

(4) results are not explained by the spatial autocorrelation of the mutual information values with respect to the underlying structural connectivity matrices;

(5) results are not driven by the species-specific SNR of functional MRI data, indicating that differences in functional MRI scanning protocols do not explain the parallel communication gap;

(6) results are not explained by the multivariate properties of the functional MRI recordings alone;

(7) results are robust with respect to functional MRI time series length;

(8) results are robust with respect to the number of subjects;

(9) results hold when contrasting parallel communication scores with respect to strict species-specific null models.

We feel that these control analyses are convincing and in line with the state-of the art literature on the topic.

I am also not sure about the information theory approach. In general, I like the idea of measuring multiple connections between nodes and going beyond the mean short path. However, I have a few questions about the method.

1) It is my understanding that the author did not use the strength of the mutual information between nodes – two nodes might have a very high or a very low mutual information, but they account them as connected as long they follow the decaying information rule.

The Reviewer is correct. Two nodes are considered to exchange information through a multi-step communication pathway if the pairwise mutual information (MI) along the structural paths is monotonically not increasing, independently from the MI value between the two end nodes of the path. This is theoretically grounded in the Data Processing Inequality theorem (Cover and Thomas, 2006).

2) The authors remove non markovian connections because the mutual information does not drop – but doesn't this mean that these nodes are actually more strongly connected (their MI is higher) ? Aren't they actually looking at the less connected nodes ?

This work focuses on relay communication, which refers to the simple transmission of information through multi-step structural paths. More complex communication mechanisms, such as feedback, convergence and information processing which may increase the information content along the path, are not modeled in our framework.

Using a simple analogy, relay communication can be thought of as the telephone (or Chinese whispers) game, where a message is whispered from person to person and then the original and final messages are compared. Although the objective is to pass the message without it becoming garbled, part of the enjoyment is that, regardless, this usually ends up happening. This is because, in a pure information-transmission system, the information content of the message cannot increase along the way; it can only remain unchanged (in the best scenario) or be degraded. Neural signaling is a noisy process involving biochemical mechanisms, so that message corruption can happen both at the ion channel and synaptic levels. Under a relay communication model, neural messages transmitted through serial white matter structural connections can therefore keep at most the same amount of information present at the source region. As also pointed out by Reviewer 3, the empirical observation in the decreasing of MI along with the physical distance of the nodes is intuitively correct and aligns well with our common sense. These considerations motivated our choice of applying the Data Processing Inequality (DPI) to explore multi-step relay communication pathways in the brain.

Please note that no brain connection is removed in our analyses. Structurally connected region pairs with strong MI are likely to occur early along a relay communication pathway, or may favor communication through their direct structural connection (i.e., through fewer alternative (and longer) structural paths).

Our DPI-based framework does not pick up noise (which one may associate to low MI values). To test this aspect, we performed extensive analyses using different null models. The null models simulate the scenarios of spatially structured and unstructured noise on top of the original structural connectome. In the first case, we randomly permuted the MI values of

individual subjects while preserving their spatial autocorrelation with respect to the underlying distance-weighted structural connectivity matrix. In the second scenario, we populated the network nodes with iid Gaussian noise with mean 0, variance 1, and the same number of time points as in experimental data (100 simulations). We assessed the number of DPI-respecting k -shortest paths in both scenarios.

The percentage of structural paths respecting the DPI in individual brain networks was significantly higher than in the null models, for all investigated species. Moreover, the distribution of group-average PCSs significantly differed from null models. Finally, we investigated whether structured and Gaussian noise may engender “communication” patterns similar to what observed in human subjects, i.e., larger parallel communication between unimodal and transmodal areas (“cross-modal” channels) compared to communication within unimodal or transmodal systems, as assessed with the DPI. We found that structured and Gaussian noise cannot explain system-dependent communication patterns. These results are illustrated in Supplementary Fig. 19 and 20.

Our analyses, which test functional brain data against different null models, show that the amount and location of DPI-respecting k -shortest paths in the brain networks of different mammalian species are meaningful and cannot be explained by structured noise alone.

3) As already mentioned by other reviewers the criterion of decreasing MI seems somewhat arbitrary. I could suggest a different criterion – find the nodes with the highest MI, and estimate the connectivity of each node based on it.

The MI is by itself a functional connectivity measure. As shown by others before us, node pairs with strong functional connectivity also tend to be structurally connected through a direct structural link, and therefore exchange information through the direct structural link. Our framework focuses on multi-step communication pathways. For this reason, we did not put any threshold on MI values and we did not remove any connection from our analyses. Importantly, our DPI results are not driven by noisy connections and significantly differ from structured and unstructured noise scenarios (please see our answer to the previous comment).

We acknowledge that DPI-based relay information transmission represents a simplistic model of brain communication and excludes multiple scenarios such as higher-order interactions (e.g., regions A and B need the mediation of region C to communicate) and processing operations (e.g., region A processes the message received from B and transmits an output to C). All these scenarios represent plausible communication models for large-scale brain networks which should be investigated in the future. In this first work, we only model relay (Markovian) information exchanges. Despite the model’s simplicity, our results highlight communication patterns that cannot be solely explained by anesthesia, brain size, white matter complexity, trivial spatial and/or topological properties of the brain networks under investigation, Gaussian noise, structured physiological noise, or spatial embedding. From a neuroscientific perspective,

these communication patterns are meaningful in that they highlight novel and well interpretable cross-species differences, brain systems' heterogeneity and individual-specific traits.

We believe that our DPI-based approach can bring a novel perspective on brain structure and function and bear meaningful neuroscientific insights. Yet, by no means we aimed at explaining the entire spectrum of communication in brain networks within this single paper. We now better highlight this point in the Discussion:

“Polysynaptic memoryless information transmission is a simple model of communication. There is no reason to assume that macroscale neural communication is limited to such a particular form. Brain network hierarchies may confer neural signals a memory of the regions previously visited along a path, thus modifying neural communication pathways in a context-dependent manner⁵⁵. This process would result in non-Markovian communication regimes. The brain may also be modeled through complex multi-object interactions not attributable to information transmission alone, such as synergistic or modulatory behaviors between multiple brain regions; feedback loops; local transformation (non-linear processing) of information^{19,40,58,59}. Biologically, these more complex communication patterns may shape important features of the mammalian brain, such as cortical temporal hierarchies^{57,58} or receptive time windows for attentional processes⁵⁹, and are worthy to investigate in future studies. Previous work showing a strong spatial heterogeneity of structure-function coupling across the cortical mantle has suggested that brain communication mechanisms may be multiplexed, with multiple protocols operating in parallel⁴⁶. The framework proposed in the present work only models Markovian information transmission and does not inform us about other complementary brain communication mechanisms. ... By no means the proposed model aims at explaining the entire spectrum of communication mechanisms in brain networks. As such, absence of relay communication (i.e., violation of the data processing inequality) may indicate absence of any communication between those particular brain regions; communication limited to one single, direct structural connection (no parallel multi-step pathways); or communication through more complex information encoding mechanisms.”

Minor comments:

There is no such thing as older species. All mammals existing today evolved over the same time period. The human brain could in theory be older than the mouse brain – unless you have evidence proving otherwise.

We agree with the Reviewer. We removed the wording 'phylogenetically older species' throughout the text.

References

Cover, T.M., Thomas, J.A. Elements of information theory (2006)

Luppi, A.I., Mediano, P.A.M., Rosas, F.E. *et al.* A synergistic core for human brain evolution and cognition. *Nature Neuroscience* **25**, 771–782 (2022)

Oh, S.W., Harris, J.A., Ng, L. *et al.* A mesoscale connectome of the mouse brain. *Nature* **508**, 207–214 (2014)

Shen, K., Bezgin, G., Schirner, M. *et al.* A macaque connectome for large-scale network simulations in TheVirtualBrain. *Scientific Data* **6**, 123 (2019)

Van den Heuvel, M.P., Bullmore, E.T., Sporns, O. Comparative Connectomics. *Trends in Cognitive Sciences* **20(5)**, 345–361 (2016)

Reviewer #1 (Remarks to the Author):

I commend the authors for their work during these multiple rounds of review.

However, after these multiple rounds of review, I think it has become clear that these results should be treated as preliminary. The current title of the manuscript reads more as a definitive claim: "Information transmission in mammalian brain networks". I would suggest a title that expresses more caution and discretion, but I leave it to the authors and editor to decide.

Reviewer #4 (Remarks to the Author):

Although I am not sure about the selection of one measurement vs. other ones, I think that the study offers many interesting ideas and results that are worth publishing and will evoke interest and discussion. So I think it is worth publishing.

REVIEWERS' COMMENTS

Reviewer #1 (Remarks to the Author):

I commend the authors for their work during these multiple rounds of review.

However, after these multiple rounds of review, I think it has become clear that these results should be treated as preliminary. The current title of the manuscript reads more as a definitive claim: "Information transmission in mammalian brain networks". I would suggest a title that expresses more caution and discretion, but I leave it to the authors and editor to decide.

We thank the Reviewer for his/her comments which helped us to improve our study. As suggested by the Reviewer and in agreement with the Editorial team, we have changed the title of the manuscript which now reads:

"Evidence for increased parallel information transmission in human brain networks compared to macaques and mice"

Reviewer #4 (Remarks to the Author):

Although I am not sure about the selection of one measurement vs. other ones, I think that the study offers many interesting ideas and results that are worth publishing and will evoke interest and discussion. So I think it is worth publishing.

We thank the Reviewer for his/her comments which helped us to improve our study.